# Neural relational inference to learn long-range allosteric interactions in proteins from molecular dynamics simulations

Jingxuan Zhu [1,2,3], Juexin Wang [2,3], Weiwei Han [1✉] & Dong Xu [2✉]

Protein allostery is a biological process facilitated by spatially long-range intra-protein communication, whereby ligand binding or amino acid change at a distant site affects the active site remotely. Molecular dynamics (MD) simulation provides a powerful computational approach to probe the allosteric effect. However, current MD simulations cannot reach the time scales of whole allosteric processes. The advent of deep learning made it possible to evaluate both spatially short and long-range communications for understanding allostery. For this purpose, we applied a neural relational inference model based on a graph neural network, which adopts an encoder-decoder architecture to simultaneously infer latent interactions for probing protein allosteric processes as dynamic networks of interacting residues. From the MD trajectories, this model successfully learned the long-range interactions and pathways that can mediate the allosteric communications between distant sites in the Pin1, SOD1, and MEK1 systems. Furthermore, the model can discover allostery-related interactions earlier in the MD simulation trajectories and predict relative free energy changes upon mutations more accurately than other methods.

[1] Key Laboratory for Molecular Enzymology and Engineering of Ministry of Education, School of Life Sciences, Jilin University, Changchun, China. [2] Department of Electrical Engineering and Computer Science, Bond Life Sciences Center, University of Missouri, Columbia, Missouri, United States. [3]These authors contributed equally: Jingxuan Zhu, Juexin Wang. ✉email: weiweihan@jlu.edu.cn; xudong@missouri.edu

Many protein functions are regulated by specific dynamic biomolecular processes, such as allostery, protein folding/unfolding, and protein activation. The biomolecular motions in these processes are primarily driven by atomic/residue interactions. Molecular dynamics (MD) simulations can directly probe biomolecular motions but may fail to capture meaningful functional information due to the limited time scale of simulation, as well as the high dimensionality and complexity of 3D trajectory data. In addition, many challenging MD analysis problems lack suitable methods to probe long-range communications. For example, allosteric communication[1,2] is well known in proteins, but understanding how signals are transmitted over long distances within a protein or across different protein molecules has been a challenging problem for decades[3,4].

Computational techniques used to model protein allosteric communication rely on graph-theoretical metrics to identify long-ranged coupling between two distal active sites. In general, a protein can be mapped to a graph, in which each node represents a residue, and each weighted edge represents an interaction between two nodes. The shortest paths between the allosteric site and the active site in a protein may be important for propagating signals in the allosteric communication. Earlier graph models used a static crystal structure to calculate the shortest paths between one residue and other residues, which may not account for the full range of potential contacts in a dynamic protein and the associated allosteric behavior[5,6]. Later, dynamic information from MD simulations was used to decipher the allosteric mechanism based on graph theory[7]. A well-known approach for allostery, perturbation response scanning (PRS)[8], uses the Hessian-based elastic network model (ENM)[8,9] to obtain correlated dynamics of positions. This model studies how the perturbation on a single residue triggers a cascade of perturbations (signals) to other nodes in an elastic network and thereby enables allosteric communication. To model the response more accurately upon ligand binding or mutation, the inverse of the Hessian has been replaced with the covariance matrix containing the dynamic properties of the system[10]. However, both models are based on the hypothetical setting, which applies the external force vector at all Cα atoms of residues in the protein. Furthermore, the assumed correlated dynamics between the allosteric and active sites may not be well detected by these methods due to the following reasons: (i) the simulation time scale may be too short to achieve sufficient signal-to-noise ratios for matrix factorizations; (ii) there may be a delay between the perturbation at the allosteric site and its response at the active site so that a linear correlation may not reflect it well; (iii) it is hard for the Hessian approach and other similar methods to differentiate between the causal and non-causal correlations related to allosteric communications. Hence, for gaining more insights into allosteric communication, it is required to develop further models.

The advent of deep learning has provided new opportunities to explore allosteric effects. The emerging graph neural network (GNN)[11] is designed to model data systems in graphs, and it has facilitated great success in solving many graph-related problems[12,13]. Recently, the GNN helped fulfill long-term research goals in modeling complex dynamic systems in traffic scenes, dynamic physical systems, and computer vision tasks by using implicit interaction models with message passing[14,15] or attention mechanisms[16]. Even more noteworthy is an unsupervised neural relational inference (NRI) model that can infer an explicit interaction structure while simultaneously predicting the dynamic model in physical simulations, such as the movement of basketball players on the court[17]. This model trains a form of variational autoencoders using motion capture data to model dynamics of the input system, in which the learned embedding (latent code) translates the underlying interaction into an interpretable graph structure and predicts time-related dynamics. NRI does not require extensive input data or prior knowledge and it does not assume any linear correlation for detecting causal relationships. For example, it successfully distinguishes whether a basketball player favors right-hand focus or left-hand focus by only depending on the state of the movement without knowledge of the underlying interactions[17].

The NRI model is suitable to learn the simulated motion trajectory of biological macromolecules in MD simulations, where biomolecules are formed by atoms connected by chemical bonds, whose motion rules are described by Newtonian mechanics. In this work, we adapted the NRI model (Fig. 1) to understand how the allosteric pathway mediates remote regulation from the ligand binding or mutation site to the active center in a protein. Based on the trajectories from MD simulations, we formulated the protein allosteric processes as the dynamic networks of interacting residues. This model uses GNN to learn the embedding of the network dynamics by minimizing the reconstruction error between the reconstructed and simulated trajectories; then, our NRI model infers edges between residues represented by latent variables. The learned embedding inherently abstracts the essential roles of the key residues in the conformational transition, which helps decipher the mechanism of protein allostery. We performed MD simulations for three allosteric systems, i.e., (i) the allosteric regulation of Pin1 induced by ligand binding, (ii) the conformational transition of SOD1 by G93A amyotrophic lateral sclerosis-linked mutation, and (iii) the activation of MEK1 by oncogenic mutations. Then, we utilized the corresponding trajectories for training the NRI model and evaluated the model performance by comparing it with three conventional approaches (constraint network analysis, derivative centrality metric of the Hessian, and dynamics coupling index). To the best of our knowledge, this study is the first attempt to use GNN, particularly NRI, to analyze MD simulations in biological systems.

## Results

**Pathways mediate inter-domain allosteric communication in Pin1**. Pin1 as an attractive therapeutic target contains an inactive N-terminal Trp-Trp (WW) domain (residues 1-39) and an enzymatically active C-terminal peptidyl-prolyl isomerase (PPIase) domain (residues 50-163) connected by a linker (residues 40-49)[18]. The PPIase domain is composed of a PPIase core (α4-helix and β4-β7 sheets), α1-α3 helices, and a semi-disordered catalytic loop (Fig. 2a). While both domains bind phospho-Ser/Thr-Pro containing substrate motifs, only the PPIase domain can isomerize the peptidyl-prolyl bond through the catalytic site[19]. Moreover, the isolated PPIase has a binding affinity that is typically 100 times weaker than the PPIase in the full-length Pin1, suggesting that the noncatalytic WW domain has the potential to remotely modulate the catalytic activity of the PPIase domain[20]. We performed two MD simulations of Pin1 in the apo and FFpSPR-bound forms[21] to evaluate the long-range effect of substrate binding to the WW domain on the flexibility of the protein backbone (Supplementary Note 1). The root-mean-square deviation (RMSD) and root-mean-square fluctuation (RMSF) values for the simulations (Fig. 2a and Supplementary Fig. 1) show that the apo form exhibits high flexibility in the WW domain (β1-β2), catalytic loop, α2-helix, and the PPIase core (β5/α4). In contrast, the flexibilities of these domains are significantly quenched when the FFpSPR binds to the WW domain, indicating that the ligand binding not only stabilizes the conformation of the WW domain but also significantly reduces the dynamic flexibility of the PPIase domain.

To explore the pathways mediating the allosteric communication by the WW domain in Pin1, we trained the NRI model on the MD trajectories with an encoder and decoder (See Supplementary Note 2 for details). Across all Pin1 ensembles of 50 sampling steps,

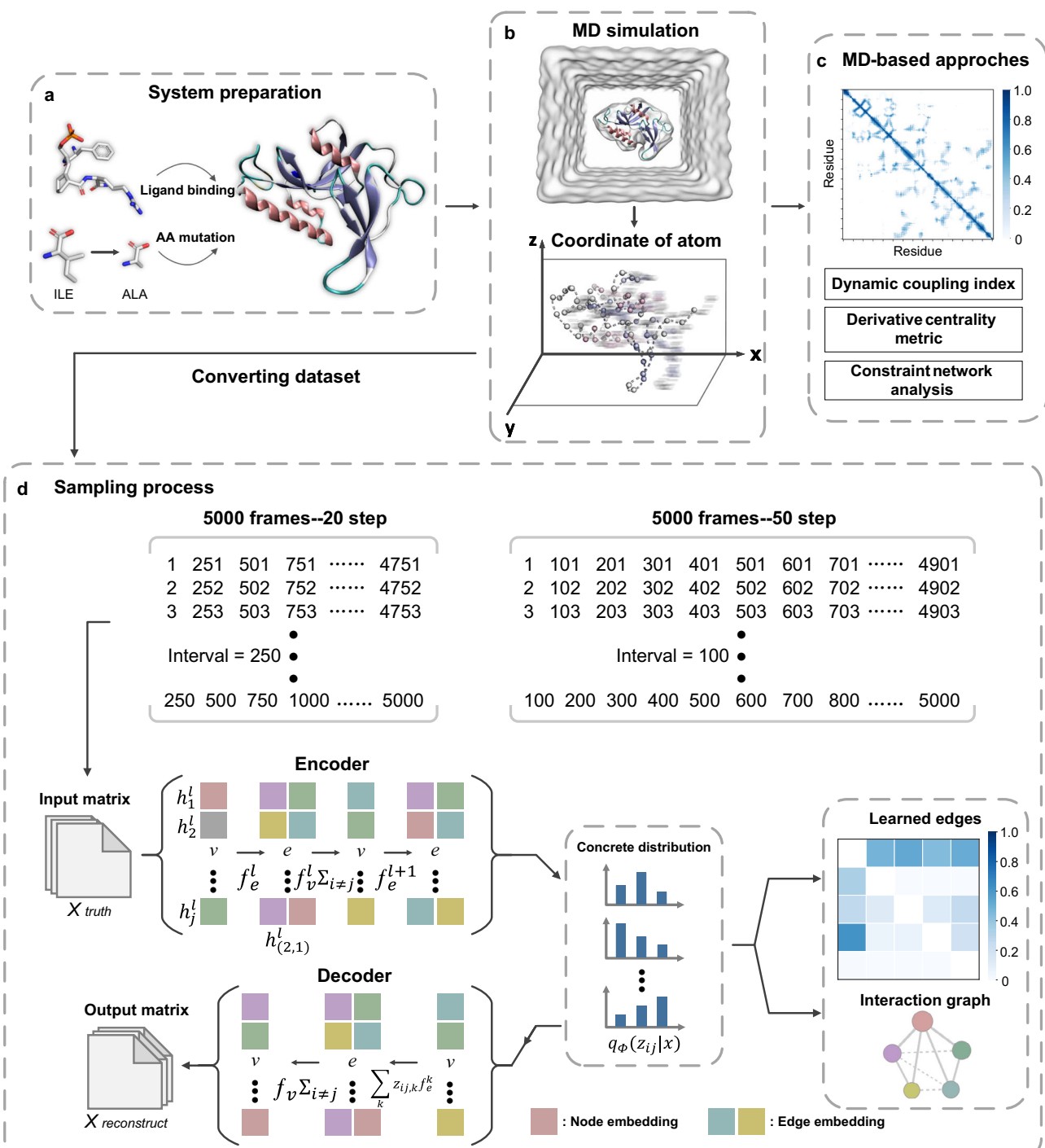

**Fig. 1 The process of inferring an interaction graph by reconstructing an MD simulation trajectory.** The process includes the system preparation of a ligand-binding complex or mutant protein structure with allostery (**a**), the MD simulation of a prepared allosteric system to obtain the trajectory with the dynamic 3D coordinates (**b**), the conventional analysis for the trajectory (**c**), and the sampling and training using the NRI model with two jointly trained components (**d**). In **d**, the NRI model consists of an encoder, which infers a factorized distribution $q_\phi(z|x)$ over the latent interactions based on the input trajectories and a decoder, which reconstructs the future trajectories of the dynamic systems given the latent graph learned from the encoder. Based on the MD trajectory, the NRI model formulates the protein allosteric process as a dynamic network of interacting residues. The interaction graph learned from this model is compared with the conventional analysis to better understand the allosteric pathway in the protein.

the model accurately reconstructs the trajectories (VSD = 0.187, 0.086, respectively) (Supplementary Fig. 2 and Supplementary Movie 1). We also obtained the distribution of learned edges between residues (Fig. 2b) as a domain interaction map (Fig. 2c) by integrating adjacent residues as blocks. The learned edges often occur between the WW domain and other domains, suggesting

that the WW domain is the key element in protein movement. Also, we calculated the shortest pathways from the residues in the WW domain to the residues in the catalytic loop based on the learned edges (Supplementary Table 1). Notably, when the FFpSPR binds to the WW domain, the correlation between the WW domain and the PPIase core is reinforced to launch the first

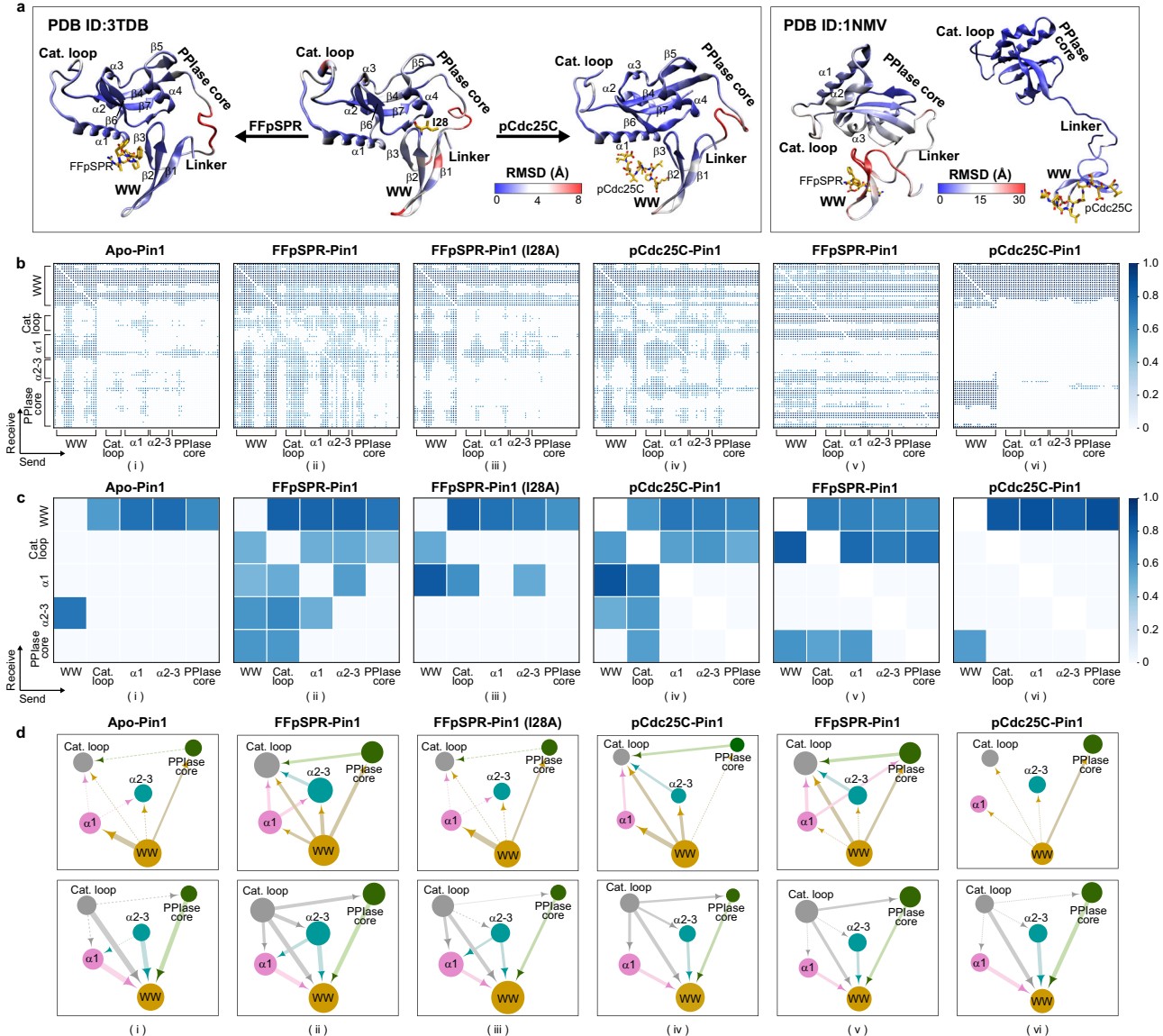

**Fig. 2 Changes in protein flexibility and interacting patterns upon ligand binding or mutation in Pin1. a** Protein flexibility of Pin1, where the color scale represents the backbone RMSD. **b** Distribution of learned edges between residues in the MD simulations of Pin1. **c** Distribution of learned edges between domains/blocks in the MD simulations of Pin1. **d** Interacting domains/blocks of Pin1, mapped from the learned edges. The Pin1 ensembles contain apo-Pin1 (PDB ID: 3TDB, i), FFpSPR-Pin1 (PDB ID: 3TDB, ii), FFpSPR-Pin1 (I28A) (PDB ID: 3TDB, iii), pCdc25C-Pin1 (PDB ID: 1PIN, iv), FFpSPR-Pin1 (PDB ID: 1NMV, v), and pCdc25C-Pin1 (PDB ID: 1NMV, vi) complexes. The size of a node represents the number of edges that directly connect to the node. The thickness of an edge represents the strength of the interaction. The arrows point to the directionality of a learned edge, i.e., the influence from the one starting domain to the ending domain. The domains/blocks presented here are WW domain (WW), catalytic loop (Cat. loop), α1-helix (α1), α2-3 helices (α2-3), and PPIase core.

two types of pathways, i.e., from the WW domain to Q131 or P133 in the PPIase core; then, the direct coupling between the PPIase core and the catalytic loop enables the allosteric communication from the WW domain to the catalytic loop via the WW-PPIase core link (Fig. 3a, left and middle).

Besides, the FFpSPR binding strengthens another communication from the WW domain via K97 in the α1-helix and S105/C113 in the α2-3 helices to the catalytic loop (Fig. 3a, right). The frequency of each residue on the paths may demonstrate the relative importance of each residue in enhancing the global connectivity and mediating capabilities that can strengthen the allosteric communication upon the substrate binding (Supplementary Fig. 3). In particular, both T29 in the interdomain interface and C113 near the catalytic site appear on the allosteric

pathways (Fig. 3a, left and right). Interestingly, the I28/T29 in the interdomain interface and C113 have been noted as vital mutation sites for impacting the activity of Pin1[22–24]. However, in the absence of ligand binding, no pathway is found from the WW domain to the catalytic loop. Although the WW domain can interact with the α1-helix, the communication cannot pass from the α1-helix to the catalytic loop (Fig. 3b and Supplementary Table 1). Thus, the ligand binding makes the WW domain and the PPIase domain more coordinated and compact to strengthen the interdomain communication in Pin1.

An NMR study[23] reported that the I28A mutation weakens interdomain interactions between the WW domain and the PPIase domain to reduce the binding affinity of the catalytic site. We simulated the I28A Pin1 of the FFpSPR-bound form[21]. The

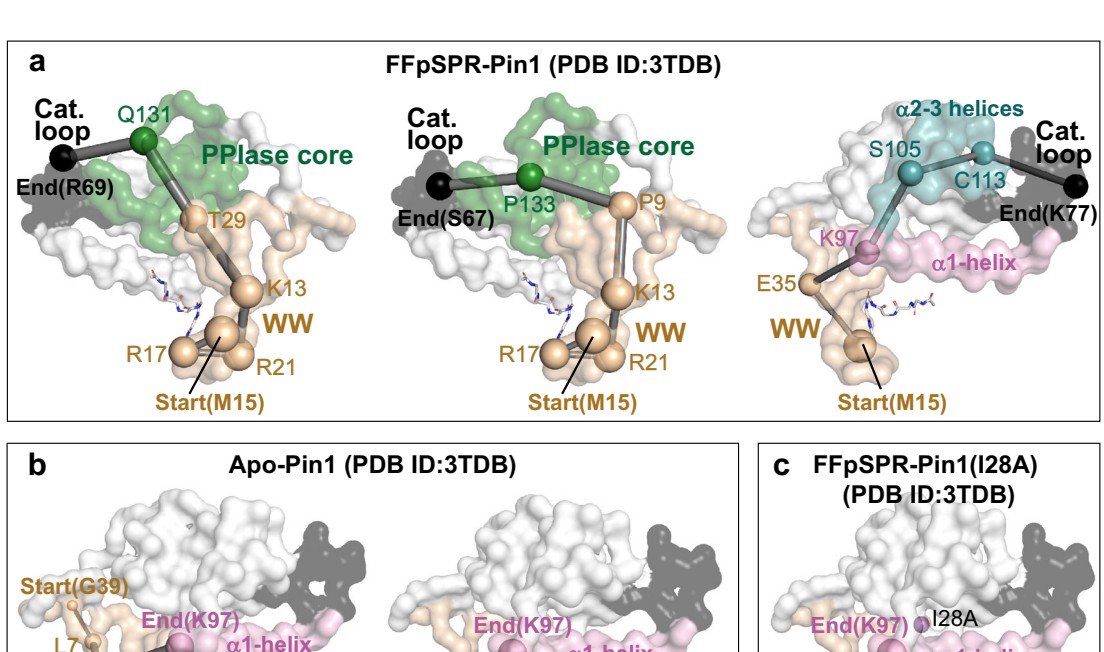

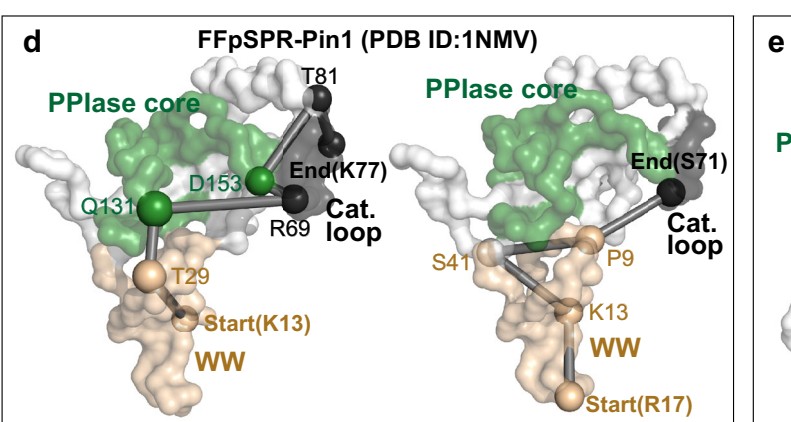

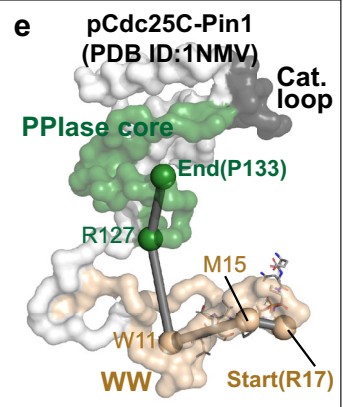

**Fig. 3 Pathways mediating inter-domain allosteric communications in Pin1, obtained from the shortest pathway calculation. a** Allosteric pathways mediate remote communication from the WW-domain (WW) to the catalytic loop in FFpSPR-Pin1. On the left, the allosteric pathway starts from the WW and continues through Q131 in the PPIase core to R69 in the catalytic loop. In the middle, the allosteric pathway starts from the WW and continues through P133 in the PPIase core to S67 in the catalytic loop. On the right, the allosteric pathway opens communication from the WW and continues through K97 in the α1-helix and the S105/C113 in the α2-3 helices to K77 in the catalytic loop. We used the residues in the WW domain as the starting point and the residues in the catalytic loop as the ending points to present the shortest pathways (additional pathways are shown in Supplementary Table 1). In **b**, the two pathways in apo-Pin1 are illustrated on the left, starting from G39, extending through L7/E35, and ending in K97 in the α1-helix. The right pathway begins at S19, then extends onward through R21/E35, and ends at K97 in the α1-helix. **c** Pathway in I28A FFpSPR-Pin1 starts in M15 and extends through R21/E35, ending at K97 in the α1-helix. **d** The two pathways in FFpSPR-Pin1 (PDB ID: 1NMV) are illustrated. On the left, starting from K13, extending through Q131/D153 in the PPIase core and ending in the catalytic loop. On the right, starting from R17, extending through K13/S41/P9 and ending in the catalytic loop. **e** One pathway that begins at R17, extends through M15/W11, and ends in the PPIase core. The size of a node represents the number of learned edges that directly connect to the node. The thickness of an edge represents the strength of the interaction. The domains/blocks presented here are WW domain (WW), catalytic loop (Cat. loop), α1-helix (α1), α2-3 helices (α2-3), and PPIase core.

trajectory's RMSF value shows that the I28A mutation increases the mobility of the whole protein structure, especially in the WW domain, the catalytic loop, and the α1-α3 helices (Supplementary Fig. 1). The learned interaction graph between key domains in Fig. 2c, d (iii) shows that the I28A mutation dramatically weakens the interactions between the WW domain and PPIase core/α2-α3 helices, which indicates that the fluctuation of the WW domain blocks the propagation of the allosteric signals from the WW to

the PPIase core and α2-α3 helices. Although the WW domain is still partially connected to the α1-helix, the α1-helix cannot bridge to the catalytic loop, resulting in the breakdown of the pathway from the WW domain to the catalytic loop via the α1-helix (Figs. 2d, iii and 3c).

The strengthened Pin1 interdomain contact upon the FFpSPR binding is referred to as positive regulation. In addition, an NMR study[25] of Pin1-WW suggests a negative regulation, i.e., a negative

allosteric peptide pCdc25C (EQPLpTPVTDL) binding to the WW domain reduces the interdomain contact, thus allowing the PPIase domain to search for a distinct pS/T-P substrate freely. To further investigate the effects of the pCdc25C binding on interdomain contact, we simulated the closed pCdc25C-Pin1 complex[21]. We found that the pCdc25C binding to WW domain of the closed Pin1 reduces the interactions between the WW domain and the PPIase core. Besides this, the edges in the networks of PPIase domain are reduced compared to the positive FFpSPR binding to the WW domain (Fig. 2b–d, iv). In addition, we used Pin1 with two domains (the WW domain and the PPIase domain) well separated (PDB 1NMV) as our starting structure to perform the simulations of the FFpSPR- /pCdc25C- bound forms[21] and trained the corresponding trajectories using the NRI model. Similarly, the model also accurately reconstructs the trajectories (VSD = 0.176, 0.139 respectively) with 50 sampling steps (Supplementary Fig. 2 and Supplementary Movie 2). From the representative conformations clustered from the trajectories (Supplementary Fig. 4), we observed that the positive allosteric ligand FFpSPR promotes an open-to-closed transition within 108 ns. However, the peptide pCdc25C binding produces a range of diverse and separate conformations. The distribution of learned edges shows that the ligand FFpSPR enables the interaction with the catalytic domain by enhancing the communication from the WW domain, through the PPIase core, and ending at the catalytic loop (Figs. 2b–d, v, and 3d, Supplementary Fig. 3). However, when the ligand pCdc25C binds, only the PPIase core interacts with the WW domain. Almost no edges connect to the catalytic loop, reflecting the reduced intradomain contacts in the PPIase domain (Figs. 2b–d, vi, and 3e).

**Allosteric effect of the G93A amyotrophic lateral sclerosis-linked mutation in SOD1.** Copper-zinc superoxide dismutase-1 (SOD1) is an oxidoreductase responsible for decomposing toxic superoxide radicals into molecular oxygen and hydrogen peroxide in two rapid steps by alternately reducing and oxidizing active-site copper[26]. The overall structure is composed of eight anti-parallel β-strands, plus two loops forming an active site (Fig. 4a). The long active loop (residues 49-83) can be divided into a dimerization loop (DL), a disulfide loop (DiL), and a zinc-binding loop (ZL). The small active loop is an electrostatic loop (EL) with residues 122-142 near the metal-binding site[27]. A study of the SOD1-linked neurodegenerative disorder amyotrophic lateral sclerosis (ALS) shows that the G93A mutation forces the EL to move away from ZL, decreasing the Zn (II) affinity of the protein[28], which affects the pathogenic process of the SOD1-linked ALS[29]. Since the G93A mutation occurs away from the metal site (Fig. 4b), this process is allosteric.

We performed MD simulations for wild type (WT) and G93A SOD1 to generate trajectories[21] for learning the interactions in SOD1 (Supplementary Note 1). The RMSF values (Supplementary Fig. 5) show that the EL of the G93A SOD1 becomes more flexible than that of the WT SOD1. Correspondingly, the motion mode reveals that the G93A mutation induces the EL far away from the metal site, while the EL of the WT SOD1 can be stabilized in the proximity of the metal site (Fig. 4b). In addition, we found that the G93A mutation makes the A93(O)-L38(N) distance increase, resulting in a decrease in hydrogen bond interaction (Supplementary Fig. 6a and Table 2). And many hydrogen-bond interactions between the β-barrel and active loops are weakened to make the G93A SOD1 structure looser than the WT SOD1 (Supplementary Figs. 6b–i and 7, and Table 2). Also, the overall dimension of the protein calculated by the radius of gyration (Rg) demonstrates a decrease in protein compactness upon the G93A mutation (Supplementary Fig. 8). To explore

how G93A mutation at the distant site significantly alters the cooperative dynamics near the active loops, we ran the NRI model on the trajectories and compared the performance of motion reconstruction. Based on the VSD values for the ground truth and reconstructed trajectories of WT and G93A SOD1 (VSD = 0.119, 0.242, respectively; see Supplementary Fig. 9), we selected the learned results of 45 steps for WT and 40 steps for G93A SOD1. As shown in Supplementary Movie 3, the reconstructed trajectory reproduces well with the simulated trajectories.

The interacting domains mapped from the learned graph show that the long active loop (DL, DiL, and ZL) and the small active loop (EL) interact with each other closely in the WT SOD1, which stabilizes the Zn (II) binding environment (Fig. 4c–e, left). A close look at the learned edges graph in Fig. 4c, left reveals that the long and small active loops also connect to the residues in the β-barrel, causing a closed EL state. Moreover, the pathways in the WT SOD1 further explain the communication pathways, starting from G93 through DL, DiL, and ZL to the EL (Fig. 4f, left and Supplementary Table 3). In contrast, during the EL opening induced by the G93A mutation, the inner connections originally in the long active loop of the WT SOD1 are almost broken, thereby loosening the network of Zn (II) binding sites (Fig. 4c–e, right). Then the allosteric pathways emanating from the A93 no longer propagate through the long active loop, but directly through the residues in the β-barrel to the EL (Fig. 4f, right, and Supplementary Table 3). Overall, the G93A mutation weakens the interaction networks within the active loops, which significantly enlarges the Zn (II) binding pocket and decreases the Zn (II) affinity with the SOD1.

**Mechanism of oncogenic mutations activating MEK1.** Mitogen-activated protein kinase (MAPKK, also known as MEK) acts as an integration point in the RAS-RAF-MEK-ERK mitogen-activated protein kinase (MAPK) signaling cascade[30]. The activation of MEK requires its phosphorylation by upstream kinases encoded by Raf oncogene[31], https://www.ncbi.nlm.nih.gov/gene/31221. The human MEK1 protein consists of a small N-terminal lobe (N-lobe) and a large C-terminal lobe (C-lobe)[32]. As shown in Fig. 5a, b, the small N-lobe is dominated by five antiparallel β-strands (core kinases domain-1) and two conserved αA/αC helices. In these two helices, the αC-helix is critical in regulating the MEK1 activity[33]. The active site of MEK1 is located at the interface of the N-lobe and C-lobe, binding to the substrate (such as ATP) or the competitive inhibitor known as A-type natriuretic peptide (ANP). The large C-lobe contains three core kinase domains, an activation segment, and a proline-rich loop. The activation segment and proline-rich loop are crucial in regulating the activation of MEK1 and downstream extracellular signal-regulated kinases (ERKs) in cells[33,34]. Recent studies reported that the E203K mutation remotely affects the active site of MEK1 to increase the phosphorylation of ERK1/2[35]. Similarly, the phosphorylation of Ser218 and Ser222 is also required for MEK1 activation to promote cell proliferation and transformation, which eventually leads to various human cancers[31].

To explore the allosteric effect of the mutation on MEK1, we performed MD simulations[21] and analyses for two nonactive MEK1s (WT and A52V)[35], two active forms (mutation E203K[35], and a phosphorylated MEK1, where both Ser218 and Ser222 are phosphorylated[31]) (Supplementary Note 1). The secondary structure changes (Supplementary Fig. 10) show that the activation segment experiences a helix-to-loop transition in the active MEK1 (E203K and phosphorylated Ser218/222). In contrast, this segment's helix content in the WT and the A52V MEK1 increased significantly compared to the active MEK1. The principal component analysis

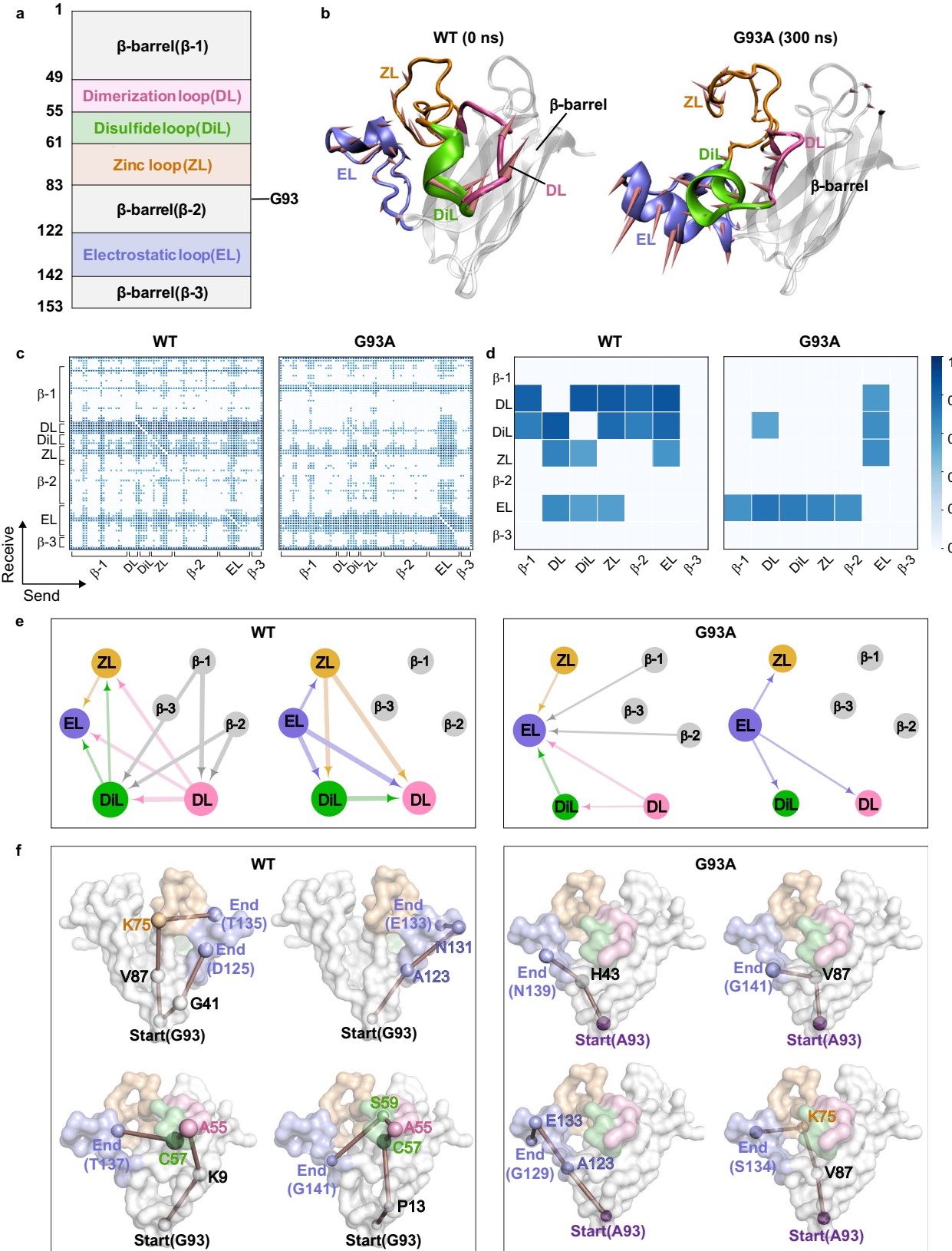

(Supplementary Fig. 11) reflects the activation segment's open trend in the active MEK1.

The above analysis only shows the changes in the dynamic motions of MEK1, which may fail to identify the common interaction features in the two active MEK1s. Thus, the NRI model was applied to learn the trajectories. The motion reconstruction results show that the reconstructions are almost consistent with the ground-truth trajectories in 60 sampling steps of WT, 90 steps of A52V, 50 steps of S218Sp/S222Sp, and 45 steps of E203K MEK1 (VSD = 0.133, 0.150, 0.183, and 0.158, respectively) (Supplementary Fig. 12 and Supplementary Movie 4). Furthermore, as shown in the learned interaction graph of nonactive MEK1 (WT and

**Fig. 4 Change of interactions between residues/domains upon G93A mutation in SOD1. a** Domain partitions of the SOD1 protein, which includes the position of the G93A mutation. **b** Initial structure of the WT SOD1 and G93A SOD1 structure at 300 ns, including a β-barrel (gray), a dimerization loop (DL colored pink), a disulfide loop (DiL colored green), a zinc-binding loop (ZL in orange), and an electrostatic loop (EL in blue). The directions shown in the graphic denote the motion mode of the protein. **c** Distribution of learned edges between residues in the MD simulations of the WT (left) and the G93A (right) for SOD1. **d** Block distribution chart of learned edges between domains in the MD simulations of the WT (left) and G93A (right) for SOD1. In **e**, the interaction graph is mapped from the learned edges for the WT (left) and G93A (right) in SOD1. The size of a node represents the number of learned edges that directly connect to the node. The thickness of an edge represents the strength of the interaction. The arrows point toward the directionality of the learned edge. In **f**, the pathways from the G93 run through the residues in the β-barrel, and the residues in the long active loop connect to the EL loop in the WT SOD1 (left); moreover, the pathways from the A93 go through the residues in the β-barrel to the EL loop in the G93A SOD1 (right). The size of a node represents the number of learned edges that directly connect to the node. The thickness of an edge represents the strength of the interaction. We used G93/A93 as the starting point and the residues in the EL as the ending points to present the pathways.

A52V) (Fig. 5c, d), few interactions occur between the domains. In contrast, the αA-helix, core kinase domain-1, activation segment, and the proline-rich loop of phosphorylated MEK1 strongly interact with other domains, which indicates that they drive the slow motion in the activation of phosphorylated MEK1 (Fig. 5c, d).

We mapped the graph of the phosphorylated MEK1 as the interacting domains (Fig. 5e, left) and calculated the allosteric pathways (Fig. 5f, left and Supplementary Table 4). Interestingly, three domains (the αA-helix, the activation segment, and the proline-rich loop) form an interaction pattern. The activation segment connects all the way to the αA-helix, which may affect the binding affinity of ANP in the active pocket. Meanwhile, the activation segment also connects to the proline-rich loop, which may activate downstream ERKs in cells. Then, we applied the NRI model to learn the inner-domain correlation from the dynamic motion of E203K MEK1. A closer look at the learned graph reveals that like the phosphorylated MEK1, the active mutation (E203K) strengthens the interactions between the activation segment/proline-rich loop and the rest of MEK1 (Fig. 5c–e). From the allosteric pathways starting with R201 (Fig. 5f, right, Supplementary Fig. 13 and Table 4), we found that the activation segment significantly affects passing messages from R201 (near E203K) to the proline-rich loop. The communication propagates through the αA-helix to the αC-helix due to the effect of the E203K mutation on the αC-helix. Hence, phosphorylated Ser218/222 and E203K mutations have a similar effect on the proline-rich loop, i.e., the activation segment as a "messenger" can interact with the proline-rich loop in their dynamics, thereby enhancing communication to the proline-rich loop.

**Effects of sampling frequency on the learned edges and their weights.** Dynamic biomolecules often undergo large-scale structural changes and visit many conformational states to perform their biological functions. This poses a problem for the NRI model training, as it is challenging to output entirely consistent learning results with different sampling frequencies. To investigate the impact of sampling frequency on the learned edges and their weights, we ran the NRI model for three case studies with 10, 15, 20, 25, 30, 40, 50, 60, 75, 90, and 100 steps for the same trajectories. Results for the MSE and VSD values are shown in Supplementary Fig. 14. The modeling with a low sampling frequency (≤ 50 steps) generally produces relatively small deviations between the ground truth and reconstructed trajectories. And the accuracy of reconstruction drops significantly with the increased sampling frequency (Supplementary Fig. 14d–f). Then, we clustered the MD trajectories using the K-means clustering algorithm. For the pCdc25C-bound Pin1 (Supplementary Fig. 15), the most populated cluster includes only 309 frames, representing 6.2% of the total processed frames. The following 20 representative clusters contain less than 200 conformations each and evenly distribute throughout the trajectory. Since the conformational

samples obtained from MD simulations are diverse and abundant, using small frames to represent the complete conformational states may be too coarse-grained and ignore too many details. For example, choosing 20 steps will result in a 250-frames interval between two consecutive steps (Fig. 1d), which indeed misses many crucial conformations vital to biological function. Hence, such a trade-off between sampling frequency and computational efficiency needs to be considered.

In addition, we showcased the distribution of learned edges for three case studies (Supplementary Figs. 16–19). Across all results with different sampling steps, the low-frequency sampling results in fewer edges and lower weights because of feeding in less structural information. Due to the discrepancy of choosing conformations in the small sampling, the edges learned are significantly different, especially for ensembles with significant conformational changes (i.e., pCdc25C-bound Pin1, Supplementary Fig. 17b). However, when the sampling is sufficient to describe a transition, the learned edges tend to be similar. Hence, we selected the learning results based on both the small reconstruction error and sufficient sampling. The results are shown in Supplementary Fig. 2 for the Pin1 study, Supplementary Fig. 9 for the SOD1 study, and Supplementary Fig. 12 for the MEK1 study. The VSD values, on average, are less than 0.2 for all three systems, which shows that the conformation distribution of the generated trajectories is almost the same as that of the actual trajectories. Although this does not suggest the exact match of atomic positions at each time point, which is not biologically meaningful in MD simulations anyway, it indicates that the statistical properties are probably the same between the ground truth and reconstructed trajectories, which is sufficient for essentially all MD simulation purposes.

To study the effects of step intervals in the NRI model learning, we learned the trajectories with different time intervals of sampling. Furthermore, we compared the RMSF values between the actual and the reconstructed trajectories, where a lower VSD value means a better model (Supplementary Figs. 20, 22, 24, and 26). It is observed that the reconstruction error slightly increases as the sampling step interval decreases when the total duration of the training trajectory keeps the same, possibly because it is harder to reconstruct trajectory details as the step interval decreases. Nevertheless, the reconstructed trajectory matches the actual trajectory relatively well even when the step interval is reduced to 5 ns or 4 ns (sampling frequency increase to 100 and 50 steps) for 500 ns' SOD1 simulation and 200 ns' Pin1 simulations (Supplementary Figs. 20 and 26). It is worth noting what step interval to use may depend on biological systems. For example, a sampling step much longer than 20 ns may be too long to recover enough information in the allosteric process (Supplementary Figs. 21, 23, 25, and 27). Our results show that a step interval of ~20 ns can yield a more reasonable outcome.

In our study, the simulations for three case studies were repeated two additional times to validate the power and accuracy of our approach. The edges learned for the three repeated trajectories

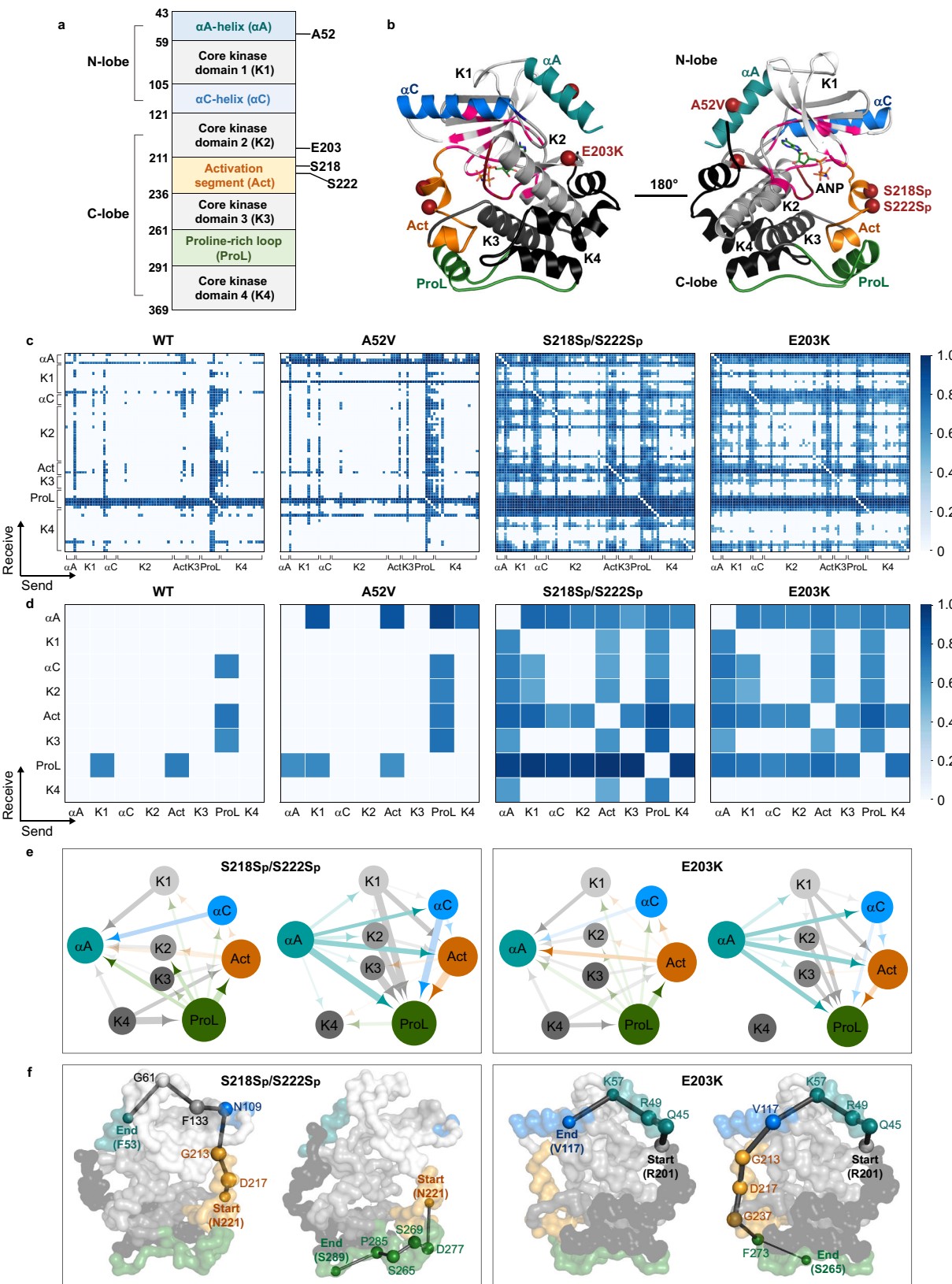

remain similar but have some differences, especially in the Pin1 and MEK1 case studies (Supplementary Fig. 28). Thus, we calculated the network node centralities (representing the importance of a residue) in allosteric pathways for the three case studies (Supplementary Figs. 29–31) and observed that the residues in the PPIase core play a crucial allosteric signal transmitting role in all three dynamic

regulations of FFpSPR-Pin1 (PDB ID: 1NMV). Upon the FFpSPR binding to the WW domain of the extended Pin1 structure, the interactions between the WW domain and the catalytic loop are supported by the edges directly connected from the WW domain to the PPIase core. Thus, the topology between the WW domain and PPIase core is stable in these repeated learning sets. Moreover, the

**Fig. 5 Changes in domain communications upon active mutations in MEK1. a** Domain partition of MEK1 protein, including the positions of mutations (A52V, S218Sp/S222Sp, and E203K). **b** Different views of the MEK1 structure. The N-terminal lobe (N-lobe) contains one core kinase (gray) and two conserved α-helices (blue). The C-terminal lobe (C-lobe) contains three core kinase domains (gray and black), an activation segment (orange), and a proline-rich loop (green). **c** Distribution of learned edges between residues in the MD simulations of WT, A52V, S218Sp/S222Sp, and E203K MEK1. **d** Distribution of learned edges between domains in the MD simulations of WT, A52V, S218Sp/S222Sp, and E203K MEK1. In **e**, the interaction graph is mapped from the learned edges of active mutant MEK1. The size of a node represents the number of learned edges that directly connect to the node. The thickness of an edge represents the strength of the interaction. The arrows denote the directionality of a learned edge. In **f**, the allosteric pathways start from N221 in the activation segment and lead to the αA-helix and the proline-rich loop in the S218Sp/S222Sp MEK1 (left). On the right, the allosteric pathways start from R201 (near E203K) and lead to the αC-helix and the proline-rich loop in the E203K MEK1. The size of a node represents the number of learned edges that directly connect to the node. The thickness of an edge represents the strength of the interaction. We used N221 and R201 (near E203K) as the starting points, and the residues in the αA/αC helices and the proline-rich loop as the ending points to present the pathways.

topologies from the WW domain to helices and from the WW domain to the PPIase core are also stable in the underlying pathways obtained from the repeats of the closed FFpSPR-Pin1 complex (Supplementary Fig. 29).

For the SOD1 system (Supplementary Fig. 30), the NRI model is also able to capture highly consistent topologies of the underlying pathways; in particular, the small active loops (DL, DiL, and ZL) stabilize the closure of the electrostatic loop. As for the A52V MEK1 study, the repeats demonstrated that the networks in the WT and A52V MEK1s are sparse compared with the MEK1 upon the active mutated (Supplementary Fig. 28). For both SOD1 and Pin1 systems, the allosteric pathways are almost learned reproducibly with fewer differences than MEK1. The difference in the edges in the MEK1 system is slightly larger. Nevertheless, the important topological elements (activation segment and proline-rich loop) are learned to illustrate the signal transmitting (Supplementary Fig. 31). Due to the chaotic and stochastic nature of molecular dynamics simulations, identical trajectories cannot be obtained even with the same set of parameters. However, the NRI model is still able to extract the key allosteric pathways related to protein dynamics regulation consistently, suggesting the model is robust.

From the methods perspective, modeling the edges explicitly is vital in the NRI architecture. To test the role of graph neural network in NRI, we performed an ablation test. We compared the proposed model and a variational autoencoder (VAE) baseline without latent variables over edges. After splitting the trajectories into training/validating/testing, the MSE results of both models on Pin1, MEK1, and SOD1 are shown in Supplementary Fig. 32 and Supplementary Table 5. We can see the latent variables over the edges can improve the model's performance, and the proposed architecture provides a better framework for modeling edges (residue interactions) of the MD trajectories than other methods. It is interesting to note that the gap between our method and the baseline is much more prominent on WT-SOD1 than on the Apo-Pin1 and WT-MEK1 systems. Supplementary Fig. 28 shows more intensive node-node interactions in the WT-SOD1 than the other two systems. Hence, the effects of graph neural network in NRI using node-node interactions over the VAE baseline (which does not consider node-node interactions) are stronger in WT-SOD1 than in the Apo-Pin1 and WT-MEK1 systems. As a result, compared to the Pin1 and MEK1 systems, the edges learned from three repeated trajectories of the SOD1 system exhibit higher consistency, indicating that the NRI model is more accurate for capturing edges in the WT-SOD1 case (Supplementary Fig. 28). This may cause a more remarkable improvement over the baseline in terms of mean squared error for WT-SOD1 than for the other two systems (Supplementary Fig. 32). It also suggests that the NRI model with latent variables over edges exhibits more significant advantages in more densely interacting systems.

**Performance comparison between the NRI model and three covariance-based models.** The long-ranged coupling between residues that gives rise to allostery in a protein is built from short-ranged physical interactions. Many computational models have been developed to predict this coupling and its allosteric relevance relies on residue-level correlations measured from MD simulations. In this section, we used the simulations of human Pin1 as a benchmark to showcase the comparisons of our NRI-based approach with three positional covariance-based methods (constraint network analysis, derivative centrality metric, and dynamics coupling index).

Based on the ensemble-based perturbation approach, constraint network analysis (CNA)[36,37] is applied to the ensembles of network topologies generated from MD trajectories for calculating the neighbor stability maps. The stability maps reflect the local stabilities of the residue-residue contacts. Hence, the contribution to free energy due to noncovalent bonding can be estimated by accumulating over the contacts in the stability map (see Supplementary Note 3 for details). This estimation cannot provide absolute free energy values, but it can show its statistical trend. Hence, we call it "free energy score". To quantitatively evaluate the performance of our model, we performed MD simulations and NRI model training for the wild type (WT) and 23 Ala-mutants of human Pin1. Given the graph learned from the NRI model, we calculated the difference of the learned edges (pairwise free energy score, $G_{Z_{ij}}$) between the WT and 23 Ala-mutants to reflect the change in structural stability caused by Ala-mutation, $\Delta G_Z = E_z^{\mathrm{Ala-mutant}} - E_z^{\mathrm{WT}}$ (Supplementary Note 6). We compared the difference of unfolding energies ($\Delta\Delta G$) between the wild type and mutants from the chemical denaturation experiments (Fig. 6a, b), with $\Delta G_Z$. The correlation between the computed free energy score ($\Delta G_Z$) based on our model and the experimental free energy ($\Delta\Delta G$) is significant ($R^2 = 0.939$, 95% confidence interval: $0.859 < R^2 < 0.974$, $p = 3.361 \times 10^{-11}$ for the interaction threshold of 12 Å; $R^2 = 0.931$, 95% confidence interval: $0.842 < R^2 < 0.971$, $p = 1.166 \times 10^{-10}$ for the interaction threshold of 15 Å) (Fig. 6c, d). In contrast, the comparison between free energy ($\Delta G_{CNA}$) computed from the constraint network analysis and experimental free energy shows a relatively poor correlation ($R^2 = 0.188$, $p = 0.390$) (Fig. 6e). Almost no correlation ($R^2 = -0.093$, $p = 0.671$) is observed between the potential energy ($\Delta G_{Total}$) from MD simulations and the experimental data (Fig. 6f).

To understand the differences of predicted interactions between methods, we identified suboptimal pathways and compared the corresponding node centralities in the pathways calculated based on the covariance matrices obtained from the CNA method and the NRI model (Supplementary Figs. 33 and 34). The node centrality heatmap in the folding stability study shows that the interactions obtained by the CNA method miss the interactions between the WW domain and the α1-3 helices, and

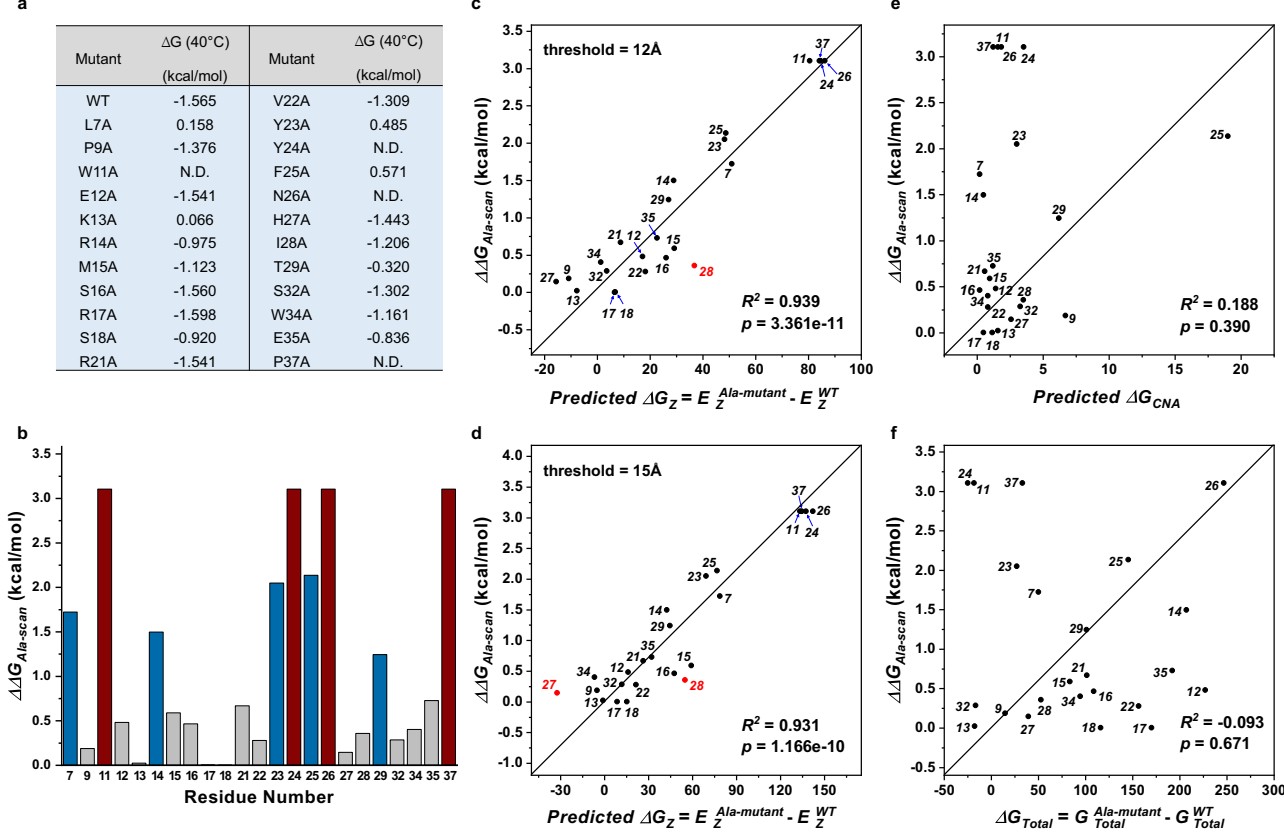

**Fig. 6 Evaluations of free energy score calculation performance of the NRI-based approach. a** Summary of thermodynamic data for the WT and 23 Ala-mutants of Pin1. N.D., not determined, represents that the mutant is too unstable to measure. **b** Effects of the Ala mutations on the equilibrium stability of Pin1. A positive value of ($\Delta\Delta G = \Delta G^{Ala-mutant} - \Delta G^{WT}$) indicates that the Ala mutation is destabilizing relative to the wild type. Mutations that destabilize more than 3 kcal/mol ($\Delta\Delta G \geq 3\ kcal/mol$) are shown as red bars, more than 1 kcal/mol and less than 3 kcal/mol ($1 \leq \Delta\Delta G \leq 3\ kcal/mol$) are shown as blue bars. **c, d** Correlation between the computed free energy score ($\Delta G_Z = E_Z^{Ala-mutant} - E_Z^{WT}$) and $\Delta\Delta G$ from the chemical denaturation experiments for the 23 Ala-mutants of Pin1. Based on the position vector of Cα in Pin1 (PDB ID: 1PIN), the threshold of residue-residue distance was set to 12 Å (**c**) and 15 Å (**d**) to present the residue-level interaction. **e** The correlation between the predicted $\Delta G_{CNA}$ (Eq. (3) in Supplementary Note 3) and $\Delta\Delta G$ for the 23 Ala-mutants of Pin1. **f** Correlation between the $\Delta G_{Total} = G_{Total}^{Ala-mutant} - G_{Total}^{WT}$ obtained from MD simulations and $\Delta\Delta G$ for the 23 Ala-mutants of Pin1. The Pearson correlation coefficient and p-value were calculated by scipy.stats.pearsonr in Python package. The p-value was computed by two-sided test and no adjustment was made for multiple comparison. The degree of freedom ($df = n - 2$) is 21.

almost all interactions concentrate between the WW domain and the PPIase core. The NRI model learning yields significantly different results. In particular, the edges between the WW domain and the α1 helix are the main interactions in the alanine mutation of positions 7, 14, 23, 25, and 29. Besides, for the structures whose alanine mutations do not affect the structural stability, the interactions are distributed from the WW domain to the α1-3 helices or the PPIase core. Because the NRI model can capture the interaction pattern changes corresponding to different structural stabilities resulting from mutations, the relative free energy scores estimated by the NRI model agree with the trend of experimental energies better than the CNA approach and MD-based method.

Based on the per-residue free energy score (per-node weight), it is possible to examine residue's (node's) importance in residue interaction networks. We validated the NRI model on the Pin1 case study: positive and negative regulation in FFpSPR- and pCdc25C- bound Pin1. The stability maps and per-residue free energy score $\Delta G_{i,CNA}$ (Eq. (4) in Supplementary Note 3) obtained using the CNA approach show that only residues in the WW domain can be identified as impactful on the structural stability upon the ligand binding (Supplementary Fig. 35a). In comparison, the interaction maps learned by the NRI-based model are more informative to allosteric communication. Specifically, the

robust nodes' weights detected in the catalytic loop, α helices, and PPIase core of the FFpSPR-Pin1 complex are vital in the allosteric pathways. However, no signal is contributed to the catalytic loop in the negative regulation due to the pCdc25C binding (Supplementary Fig. 35b).

Further, we compared the application of the NRI model with the derivative centrality metric[38]. Derivative node metric ($\delta_{node}$)[38] for the FFpSPR- and pCdc25C- bound Pin1 identifies the node's importance at conveying the covariance between sources (WW domain) and sinks (catalytic sites) (Supplementary Note 4). First, we showcased the distribution of learned edges for frames 1–1000, 1000–2000, …, 4000–5000 of the trajectories (Supplementary Fig. 36). It shows that the dynamics of biomolecular change considerably over time. In contrast, the distributions in frames 1–500, 1–1000, …, 1–5000 of the trajectories remain relatively stable (Supplementary Fig. 37), which may reflect the overall features of the whole dynamic process, instead of each segment in the process. Hence, we will use the distribution of frames 1-N in the following analyses.

As presented in Fig. 7a, the catalytic sites in the FFpSPR-Pin1 exhibit large $\delta_{node}$ values after 200 ns (frame 2000). Thus, the complete allostery propagation is detected after 200 ns (frame 2000) by measuring the derivative centrality metric. In comparison, the NRI approach has the potential to capture an allosteric signal transmitted

from the WW domain to the catalytic loop in a shorter simulation time. The relatively large edge weights in the catalytic loop are learned within 50 ns (frames 1–500) (Fig. 7c). The corresponding conformational changes show that the open conformation completes the closing transition at ~108 ns after FFpSPR binding to the WW domain of Pin1 (Fig. 7c and Supplementary Fig. 4a). Hence, the NRI approach captures the allosteric signal propagated to the catalytic sites before observing the complete open-to-closed transition.

Then, the Pearson correlation coefficient ($R^2$) between the learned node's weight and the residue-level RMSD value, on average, more than 0.7, demonstrates that the strength of the learned interactions is associated well with the structural stability changes (Supplementary Figs. 36c, d, 38). By mapping representative conformation with the average RMSD value in trajectories of different time scales (Fig. 7c), we found that from 50 ns to 100 ns (frames 500–1000), the RMSD value in the catalytic loop of FFpSPR-bound Pin1 increased by 3 Å, reflecting the reinforced weights in edges connecting to the catalytic sites. Then, as the simulation time increased to 200 ns (frame 2000), the structural destabilization in α2-α3 helices represents the enhanced interaction weights. In the later simulation, the stabilities of the catalytic loop and α2-α3 helices do not change much over time, so that the weights of the learned edges no longer increase significantly. Since the conformation remains open for the simulation of pCdc25C-bound Pin1, almost no signal could be transmitted to the catalytic loop (Fig. 7b, d).

Moreover, we compared the NRI model with the dynamic coupling index (DCI)[10], which replaces the Hessian matrix with the covariance matrix obtained from MD simulations to model changes in interaction networks upon ligand binding more accurately (Supplementary Note 5). The DCI metric identifies the dynamic allosteric residue coupling between two distal residues/domains based on such a matrix. We compared the DCI profile between the WW domain and the rest of Pin1 to the nodes' weights learned from our NRI model. After 100 ns (frame 1000), the DCI metric presents the strong coupling from the WW domain to the catalytic loop (Supplementary Fig. 39a). Our model is sensitive enough to detect the signal conveying to the catalytic loop in the first 50 ns (frames 1–500) as indicated in the more robust weights in the catalytic loop than other domains in Pin1 (Supplementary Fig. 39b).

Finally, we compared the allosteric pathways learned from the NRI model and the other methods. For the FFpSPR-Pin1 complex in the Pin1 case study, unlike the result of the NRI model, the distributions of edges learned from DCI, Hessian, and CNA methods do not contain the expected interactions between the WW domain and the α1-3 helices (Supplementary Figs. 40 and 41). Interestingly, D112A/N and C113S caused a considerable reduction in the catalytic activity. E100D, E104K, S105F, S106*, and D112N have been reported as somatic mutations in cancer[24,39]. Thus, mutating these residues in the α1-3 helices may cause chemical shift perturbations in the interaction between the catalytic loop and the PPIase domain. The NRI model successfully learns the interaction patterns in the α2-3 helices that are ignored by the other three methods. Similarly, upon the pCdc25C binding to the WW domain of the closed conformation, the edges learned from the NRI model are distributed between the WW domain and the helices in the PPIase domain; other methods tend to capture the interaction from the WW domain to the PPIase core. For the extended conformation of FFpSPR-Pin1, the NRI model, Hessian, and CNA methods all capture the high node centralities in the PPIase core. In contrast, the DCI method does not capture the topology from the WW domain to the PPIase core. This suggests that the NRI model may be more consistent with the positively and negatively controlled allosteric regulation than other methods.

For the WT SOD1 case study, in contrast to the NRI model, the Hessian and CNA methods ignored the pathways from the DL, DiL, and ZL to the EL; DCI method missed the pathway from the DL and DiL to the EL (Supplementary Figs. 42 and 43). This contradicts the observation that the long active loop (contained by a DL, DiL, and a ZL) is crucial in stabilizing $Zn^{2+}$ binding to the active sites[27]. For the MEK1 case study, the distribution of edges learned from the NRI model and the other three methods are similar (Supplementary Figs. 44 and 45). Except the pathways directly starting from the activation segment to the αC helix in the S218Sp/S222Sp MEK1, the NRI model determines that the proline-rich loop also plays a bridging role in this message passing. Notably, the proline-rich loop activates the downstream ERKs in cells[33,34].

In summary, the results for comparison demonstrate that (i) computed relative free energy scores using our NRI-based approach agree very well with experimental data, and (ii) the NRI model can effectively capture residue-level interactions within a shorter simulation time.

## Discussion

This study applied a GNN-based NRI model to analyze latent interactions between residues from reconstructing MD trajectories of proteins. We carried out three case studies to explore the allosteric long-range interactions for the Pin1, SOD1, and MEK1 systems. We have demonstrated that our NRI model can effectively generate the interaction graphs related to the protein's slow motion by embedding reconstructed MD trajectories. The shortest pathways between the allosteric site and the active site in the interaction graphs can reveal the pathways mediating allosteric communications. In addition, the model can capture allostery-related interactions and the trend of dynamic motions using a shorter simulation than other methods.

Recently, two distinct allosteric mechanisms of human Pin1 have been well studied using computational approaches. Gratifyingly, we have some findings consistent with some recent result[40], which shows that the presence of the positive allosteric ligand FFpSPR enhances the interdomain interaction between the WW domain and the PPIase domain through two pathways. Path1 emanates from the WW backside and propagates through the inter-domain interface and the PPIase core to the catalytic sites; Path2 emanates from the WW front pocket and propagates through the bound substrate, α1, and the α1-core interface to the catalytic loop. Our results not only show the strengthened interdomain contact but also identified another pathway of open communication from the WW through the α1 and α2-3 helices, ending in the catalytic loop (see Fig. 3a, right). In addition, we investigated the effects of the positive and negative ligand on the conformational transition of the well-separate structure. The result on the negative mechanism confirms a recent finding, i.e., the negative allosteric ligand pCdc25C binding reduces the intradomain contact in the PPIase domain.

The allosteric pathways derived from the shortest paths provide valuable information when considering protein design. It may be possible to mutate the residues in the allosteric pathways to alter the biological functions and regulatory properties of proteins. One example we demonstrated is residue I28 in Pin1 with a known impact of allostery appearing next to T29. The exemplary residue T29 is a key residue on one pathway identified in Fig. 3a. Furthermore, both residues R49 and K57 in the αA-helix appear on the allosteric pathways of the two activated MEK1s (Supplementary Fig. 13). Since the αA-helix is the critical interface interacting with the rest of the kinase domain, the mutations of residues R49 and K57 are likely to cause significant alterations in the helical structure, thereby inducing ERK

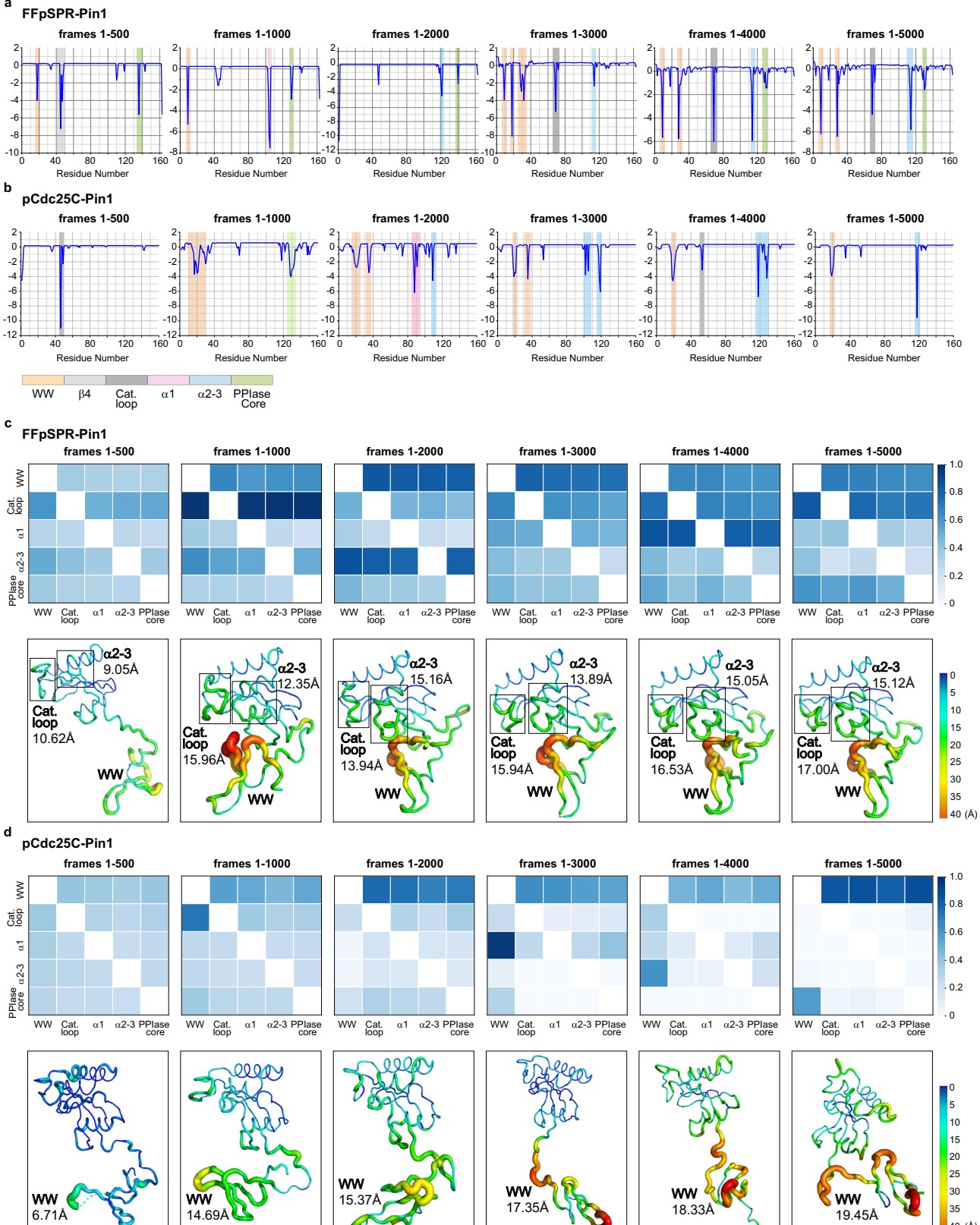

**Fig. 7 Comparison of the performance of the Hessian-based and NRI-based approaches in capturing the allosteric signals in simulations. a, b** Derivative node metrics $\delta_{node}$ based on Hessian as a function of residues for FFpSPR- and pCdc25C-bound Pin1 systems. The metrics are calculated using frames 1–500, 1–1000, ..., 1–5000 of the trajectories. Residues that have strong connections are highlighted by vertical bars, whose colors depend on the domains. **c, d** Distribution of learned edges between the domains and the corresponding average conformation mapped with RMSD values for FFpSPR- and pCdc25C-bound Pin1 system. The edges are learned from frames 1–500, 1–1000, ..., 1–5000 of the trajectories. The domains presented here are WW domain (WW), catalytic loop (Cat. loop), α1-helix (α1), α2-3 helices (α2-3), and PPIase core.

phosphorylation[41]. In addition, residues G213 and D217 in the activation segment show a significantly high frequency on the pathways of the two activated MEK1s, which confirms their roles as global mediating sites in allosteric communication (Supplementary Fig. 13). Mutations in this region indeed lead to constitutive activation of the MAPK pathway[42]. Thus, the allosteric pathways learned by our NRI model may potentially reduce the need for mutation screening significantly by a targeted design. Such an approach can be used for general mutation effect predictions, as well as a guide in designing allosteric drugs capable of modulating protein function with potentially higher specificity and lower toxicity than traditional drugs.

Due to large energy barriers, the conformational changes of biomolecules usually occur in milliseconds or longer time scales, which is typically inaccessible in the MD simulations with hundreds of nanoseconds to tens of microseconds. On the other hand, the driving forces leading to the long-term conformational changes and the underlying inter-/intra-domain interactions reveal themselves long before the conformational changes are revealed in trajectories. As reported, many analysis methods of MD simulations have been used to identify allosteric importance based on positional covariance-based metrics. Our study performed a range of comparison studies between our NRI-based model and other MD approaches (constraint network analysis, derivative centrality metric of the Hessian, and dynamics coupling index). The current NRI models indeed take more time to compute than the other methods compared. Nevertheless, the strength of the NRI model is not the computing time but rather its potential to identify some long-range interactions that other methods may miss. This does not mean a replacement of other methods, but our method is complementary to them. The advantages of our model are as follows: (i) our model is sensitive enough to capture the allosteric signal earlier before observing the complete conformational transition; (ii) our model can clearly present the interaction patterns and signal-transmit pathways during protein biological function; and (iii) our model has significant potentials in estimating free energy changes upon mutations.

This model is not restricted to allosteric regulation. Many other biological and pharmaceutical processes, such as protein folding/ unfolding, protein activation, or drug molecule binding targets can also be formulated as a dynamic interaction graph by the NRI model. In particular, the NRI model is appealing when probing non-periodic biomolecular motion. Unlike the periodic physical movement where interactions do not change over time, proteins during performing functions are often accompanied by considerable conformation and interaction changes. Using NRI in those cases will retrieve interactions over time. We believe the NRI model can be developed to recover the interactions between residues at every time interval in the process of performing protein functions. Additional NRI methods, such as dynamic NRI[43] can be applied for this purpose. Future studies include making this model more robust, computationally more efficient, and biologically more explainable, which will lead to a useful software tool for analyzing MD trajectories in general.

## Methods

**NRI model**. The NRI model[17] consists of two co-training parts: an encoder to predict the interaction given the dynamic system's trajectories, and a decoder to predict the trajectories of the dynamic system given in the interaction graph. Specifically, the input consists of $N$ nodes. The feature vector (position and velocity in the dimensions of x, y, and z) of node $i$ (input/output dimension of 6 for each node) is denoted as $x_i^t$ at time $t$. All $N$ nodes' feature set is denoted as $x^t = \{x_1^t, \cdots, x_N^t\}$. The trajectory of node $i$ is denoted as $x_i = \{x_i^1, \cdots, x_i^T\}$, where $T$ is the number of time steps. Finally, all trajectory data are recorded as $x = \{x^1, \cdots, x^T\}$. The NRI model simultaneously learns the edge values and reconstructs the future trajectories of the dynamic system in an unsupervised

manner based on an unknown graph $z$. The interactions between nodes $i$ and $j$ take the form of a latent variable $z_{i,j} \in \{1, ..., K\}$, in which $K$ is the number of interaction types being modeled. These interaction types do not have any pre-defined meaning, but rather the model learns to assign a meaning to each type. The structure of the model is presented in Fig. 1. The code is freely available at https://github.com/juexinwang/NRI-MD.

This model is formalized as a variational autoencoder (VAE)[44,45] that maximizes the evidence lower bound (ELBO):

$$\mathcal{L}(\Phi, \theta) = \mathbb{E}_{q_{\Phi(z|x)}}[\log p_\theta(x|z)] - KL[q_\Phi(z|x)||p_\theta(z)] \quad (1)$$

in which $\Phi$ and $\theta$ are trainable parameters of probability distributions. This formulation consists of three primary probability distributions: First, the encoder $q_\Phi(z|x)$, returns a factorized distribution of $z_{i,j}$, in which a one-hot encoding representation of the $K$ interaction types is used on $z_{i,j}$. For easy visualization, we used $K=4$ types to train the model. The first edge type is "hard-coded" as nonedge and was trained with a prior probability of 0.91. All other edge types received a prior of 0.03 to favor sparse graphs. These priors were the default values in the original NRI model[17]. Second, the decoder $p_\theta(x|z)$, reconstructs the dynamic systems given a sampled set of interactions $z_{i,j}$. Third, the prior $p_\theta(z)$, is a uniform independent categorical distribution per relation variable.

More formally, as shown in Fig. 1d (on the left), the encoder takes the form

$$q_\Phi(z_{ij}|x) = softmax(f_{enc,\Phi}(x)_{ij,1:K}) \quad (2)$$

in which $f_{enc,\Phi}(x)$ is a GNN performed on the fully connected networks (without self-connection) to predict the latent graph structure. The encoder operation is presented as follows:

$$h_j^1 = f_{emb}(x_j) \quad (3)$$

$$h_{(i,j)}^l = f_e^l\left(\left[h_i^l, h_j^l\right]\right) \quad (4)$$

$$h_j^{l+1} = f_v^l\left(\sum_{i \neq j} h_{(i,j)}^l\right) \quad (5)$$

$$h_{(i,j)}^{l+1} = f_e^{l+1}\left(\left[h_i^{l+1}, h_j^{l+1}\right]\right) \quad (6)$$

where $h_j^1$ is the embedding of node $v_i$ in layer $l$, $h_{(i,j)}^l$ is an embedding of the edge $e_{(i,j)}$. Equations (4), (5) represent node-to-edge ($v \rightarrow e$) and edge-to-node ($e \rightarrow v$) operations, respectively. The encoder runs two rounds of node-to-edge ($v \rightarrow e$) and an edge-to-node ($e \rightarrow v$) message passing. The node-to-edge operation generates the edge features connecting the node embeddings and the edge-to-node operation aggregates the message of edge embeddings from all incoming edges. Since the graph is fully connected, each node obtains a message from the entire graph. Finally, all messages pass from nodes to edges. In our implementation model, every message passing operation is performed by a 2-layer perceptron[17].

The distribution of $z$, $q_\Phi(z|x)$, is learned from the encoder. Then the sampling is performed to generate $z_{ij}$ only available in the $K$ edge type. We sampled from a continuous approximation of the discrete distribution and used reparameterization to obtain gradients from this approximation, which were calculated as[46]:

$$z_{i,j} = softmax((h_{(i,j)}^2 + g)/\tau) \quad (7)$$

where $g \in \mathbb{R}^K$ is an independent and uniformly distributed vector from the Gumbel distribution (0, 1), and $\tau$ (softmax temperature) represents the smoothness of sampling. The distribution tends to become one-hot encoded samples when $\tau \rightarrow 0$.

The decoder is expressed as:

$$p_\theta(x|z) = \prod_{t=1}^T p_\theta(x^{t+1}|x^t, \cdots, x^1, z) \quad (8)$$

which reconstructs the dynamic systems $p_\theta(x^{t+1}|x^t, \cdots, x^1, z)$ with a GNN given the latent graph structure $z$. A recurrent decoder with a GRU unit[47] is required to model $p_\theta(x^{t+1}|x^t, \cdots, x^1, z)$. The decoder operation is presented as follows:

$$\tilde{h}_{(i,j)}^t = \sum_k z_{ij,k} \tilde{f}_e^k([\tilde{h}_i^t, \tilde{h}_j^t]) \quad (9)$$

$$MSG_j^t = \sum_{i \neq j} \tilde{h}_{(i,j)}^t \quad (10)$$

$$\tilde{h}_j^{t+1} = GRU\left(\left[MSG_j^t, x_j^t\right], \tilde{h}_j^t\right) \quad (11)$$

$$\mu_j^{t+1} = x_j^t + f_{out}\left(\tilde{h}_j^{t+1}\right) \quad (12)$$

$$p(x^{t+1}|x^t, z) = \mathcal{N}(\mu^{t+1}, \sigma^2 \mathbf{I}) \quad (13)$$

in which $z_{ij,k}$ is the $k$-th element of the vector $z_{ij}$, $\sigma^2$ is a fixed variance, $x_j^t$ is the correct input, $\mu_j^t$ is the predicted mean, and $f_{out}$ denotes an output transformation. The decoder runs multiple GNNs in parallel to the encoder. In the node-to-edge

($v \rightarrow e$) message passing, Eq. (9), the input is the recurrent hidden state from the previous time step. The hidden state of an edge is determined by the hidden state of its connecting nodes, and it allows the message at each time step to pass through the hidden state. Thus, the prediction at $t + 1$ is based not only on the previous time step but also on messages from all the previous time steps. In the edge-to-node ($e \rightarrow v$) message passing, shown in Eqs. (10)-(12), the concatenation of the aggregated messages $MSG_j^{t+1}$, the current input $x_j^{t+1}$, and the previously hidden state $\bar{h}_j^t$, is denoted as the input of node $v_j$ into the GRU to generate the hidden state at the next time step. Then, the value observed previously and the hidden state at the current time step are used to predict the state's distribution (position and velocity) in future time steps.

The ELBO described in Eq. (1) has two terms: first, the reconstruction error $\mathbb{E}_{q_{\Phi(z|x)}}[log\, p_\theta(x|z)]$, which assumes the predicted outputs represent means of a Gaussian distribution with the fixed variance $\sigma$ and is calculated through:

$$\mathbb{E}_{q_{\Phi(z|x)}}[log\, p_\theta(x|z)] = -\sum_j \sum_{t=2}^T \frac{||x_j^t - \mu_j^t||^2}{2\sigma^2} + const \quad (14)$$

Second, the KL divergence $KL[q_\Phi(z|x)||p_\theta(z)]$, is the sum of entropies and a constant:

$$KL[q_\Phi(z|x)||p_\theta(z)] = \sum_{i \neq j} H(q_\Phi(z_{ij}|x)) + const \quad (15)$$

where $H$ represents the entropy function. The constant term is due to the uniform prior, which leads to marginalization of one of the encoder terms in the loss.

The whole training process was carried out as follows: (i) We first performed the encoder to calculate $q_\Phi(z_{ij}|x)$ given a training MD trajectory X; (ii) we then sampled $z_{ij}$ from a continuous approximation of the discrete distribution, and (iii) we finally ran the decoder to reconstruct the interacting dynamics $p_\theta(x^{t+1}|x^t, \cdots, x^1, z)$ for the Pin1, SOD1 and MEK1 systems.

**Software implementation**. Tools and packages used in this paper include: Python version 3.7.1, torch version 1.2, matplotlib version 3.1.1, seaborn version 0.9.0, numpy version 1.19.5, networkx version 2.3, argparse version 1.1, pandas version 0.25.1, R version 3.6.2, CNA version 2.0, VMD 1.9.3, AmberTools16, CPPTRAJ version 16.16, Cytoscape version 3.8.0, Autodock 4.2, and SWISS-MODEL server (https://swissmodel.expasy.org/).

**Reporting summary**. Further information on research design is available in the Nature Research Reporting Summary linked to this article.

## Data availability

The data that support this study are available from the corresponding authors upon reasonable request. Three case studies data sets have been deposited in the Zenodo database (https://doi.org/10.5281/zenodo.5941385). The initial structures of Pin1 system obtained from PDB 3TDB, 1PIN, and 1NMV. The initial structures of SOD1 system obtained from PDB 2C9V. The initial structures of MEK1 system obtained from PDB 3SLS. Source data are provided with this paper.

## Code availability

The tool described in this study is open source and publicly available at GitHub (https://github.com/juexinwang/NRI-MD).

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

## Acknowledgements

This work was supported by the China Scholarship Council [201906170166] to J.Z., and the Overseas Cooperation Project of Jilin Province [20200801069GH] to W.H., and the National Institute of General Medical Sciences of the National Institutes of Health [R35-GM126985] to D.X. We also thank Ms. Carla Roberts for thoroughly proofreading this paper.

## Author contributions

Conceptualization: D.X. and W.H., methodology: J.Z., J.W., and D.X., software coding: J.W. and J.Z., data simulation and collection: J.Z., data analysis: J.Z. and J.W., software testing and tutorial: J.W. and J.Z., manuscript writing, review, and editing: J.Z., J.W., W.H., and D.X.

## Competing interests

The authors declare no competing interests.
