## [Peer Review File · Nature Communications]

Neural relational inference to learn long-range allosteric interactions in proteins from molecular dynamics simulationsReviewers' Comments:

Reviewer #1:

Remarks to the Author:

Zhu et al., we developed a neural relational inference (NRI) model based on a graph neural network (GNN) approach to capture dynamic network residues modulating allosteric interactions. The model trains a form of variational autoencoders using 50 uniform time steps sampled at 4ns second intervals of 200 ns Molecular Dynamics (MD) trajectory as ground truth to model dynamics of the protein system, where the learned embedding (latent code) translates underlying dynamic network of interaction into an graph structure and also predicts the time related dynamics. They applied their approach to three different system where allosteric interaction play critical role: (i) allosteric regulation of PIN1 by binding of a ligand to non-catalytic WW domain, (ii) allosteric G93A mutation of SOD1 and (iii) oncogenic mutation modulating MEK1. The authors showed that RMSF and MSE values that are obtained from reconstructed trajectories captures those obtained directly from MD trajectories. They also use obtained GNN to provide mechanistic insights about allosteric regulations in these three systems. This is a novel approach and applications of three different systems also show the strength of the approach. However, the method needs more testing and detailed comparison with the other MD based methods in the field before it gets published. Only then the main and the most important conclusion, that is their NRI based model can capture the interaction patterns which cannot be obtained by other MD method, can be justified. The manuscript should be revised by addressing the issues discussed below

(i) It is not clear how their GNN approach is better in capturing non-linear correlation that cannot be capture with co-variances matrices or cross-correlations maps of MD analysis. The findings obtained from GNN should compared with other MD based approaches to show that it is indeed novel, and capture non-linear correlations as stated. (See related allostery works by HX Zhou, Barahona, McCullagh, Ozkan, Gohlke)

(ii) There is also lack of discussion/comparison that are specific to these three systems. All of the three systems have been extensively studied by other computational groups. The authors should add a discussion whether their results overlap with those findings, or whether their results differ and possible discussion about the sources of difference and overlaps.

(iii) In same context, the identified results are not compared or discussed with the available experimental data. For example, NMR analysis of PIN1- WW suggest a negatively controlled allosteric regulation where substrate binding of WW domain a reduces the interdomain contact as compared to that of the Apo state, thus allowing the PPIase domain to freely search for a distinct pS/T-P substrate. This plausible mechanism suggested by the results of NMR analysis does not completely agree with the findings of GNN analysis of the present work. Why is that ? This brings the issue of the approach and justification of simulation and sampling.

(iv) In their approach, the training of NRI (their learning system) depends on sampling of the trajectories (basically output frequency). There is no justification of choosing 50 frames. They should add a section where they study impact of sampling frequency on the edges learned and their weight. What is the uncertainty on those edges and their weights ? One would expect to see variation in long distance coupling with lower sampling frequency

(v) This as brings up another point of simulation time. The authors used 200 ns simulations after the equilibration. I recommend increasing the simulation time to 500 to 1 micro seconds, and will the results obtained from longer trajectories still capture the same long-range coupling and pathways emerged from the 200ns? Will the weights of edges, connecting different domains remain same ? This analysis is necessary to show the power and accuracy of the approach.

Reviewer #2:
Remarks to the Author:
Key Results:

This paper applies the neural relational inference (NRI) model to learn the latent interactions among residues of proteins given the molecular dynamics (MD) trajectories. The idea is to treat the protein as a graph where each node is a residue, each edge represents the interaction between two residues, and the node feature includes the 3D velocity and 3D position. The NRI model is trained to reconstruct the given MD trajectories of proteins.

Significance:

I am incompetent in judging the significance of this paper to the related biological fields. From the perspective of machine learning (ML), this paper basically applies the neural relational inference model proposed in [1] to learn the latent interaction among residues of proteins. It is a novel and useful application of this ML model.

[1] Kipf, T., Fetaya, E., Wang, K.C., Welling, M. and Zemel, R., 2018, July. Neural relational inference for interacting systems. In International Conference on Machine Learning (pp. 2688-2697). PMLR.

Data and Methodology:

The data and non-machine-learning part of the methodology are outside the scope of my expertise. From the perspective of machine learning, the design of applying the NRI model to solve the problem is technically sound, i.e., treating the protein as a graph and use NRI to learn the latent interactions among residues as well as reconstructing molecular dynamics trajectories. Once trained, the decoder along with the fixed prior of the NRI constructs a generative model. I am curious how good the generated trajectories are from this generative model. Are there any useful applications for this generative model in the current problem context?

Validity & Analytical Approach:

From the perspective of machine learning, I have two concerns regarding the evaluation.

1, It is mentioned that traditional ML methods like principal component analysis (PCA) and cross-correlation analysis (CCA) were applied to this problem in the literature. They are restrictive due to their linearity. However, I did not find any comparison of NRI versus these traditional methods in any form. It would be necessary to include such comparisons since one of the main motivations is that NRI can model nonlinear correlation in dynamics whereas traditional methods can not.

2, Since the main goal of the proposed model is to learn the latent interaction of residues, it is necessary to quantitatively measure how accurate these learned latent interactions are. I am not sure whether authors could obtain the ground-truth values of these latent variables for a subset of data. If that is not the case, it would still be necessary to obtain some indirect quantitative metrics. I did not find any quantitative metrics in the current draft.

Suggested Improvements:

Please see the other comments for specific suggestions.

Clarity and Context:

1, The decoder part is not clearly explained. In particular, from Eq. (5), it seems that the autoregressive conditional probability on the right-hand side of Eq. (3) is a Gaussian. However, it is not explicitly mentioned anywhere what kind of distribution is being used. From line 403 to line 413, it is unclear how the recurrent update and the GNN are mixed to implement this autoregressive conditional probability. Furthermore, Eq.(5) is not accurate in describing the dependency on the latent variable z since z does not appear on the right-hand side. It would be better to clearly explain how μ and σ depend on z .

2, Some mathematical notations need to be improved and/or clarified. For example, the subscript of the encoder used in Eq. (1) is very confusing and not explained anywhere.

3, It would be great to clearly explain what the $K=4$ edge types are in line 394.

4, What is the exact performance metric used during the validation to determine the best hyperparameters?

References:

It would be great to cite the original GNN paper [2] since GNN has been mentioned many times throughout the paper.

[2] Scarselli, F., Gori, M., Tsoi, A.C., Hagenbuchner, M. and Monfardini, G., 2008. The graph neural network model. *IEEE transactions on neural networks*, 20(1), pp.61-80.

REVIEWER COMMENTS

Reviewer #1 (Remarks to the Author):

Zhu et al., we developed a neural relational inference (NRI) model based on a graph neural network (GNN) approach to capture dynamic network residues modulating allosteric interactions. The model trains a form of variational autoencoders using 50 uniform time steps sampled at 4ns second intervals of 200 ns Molecular Dynamics (MD) trajectory as ground truth to model dynamics of the protein system, where the learned embedding (latent code) translates underlying dynamic network of interaction into an graph structure and also predicts the time related dynamics. They applied their approach to three different system where allosteric interaction play critical role: (i) allosteric regulation of PIN1 by binding of a ligand to non-catalytic WW domain, (ii) allosteric G93A mutation of SOD1 and (iii) oncogenic mutation modulating MEK1. The authors showed that RMSF and MSE values that are obtained from reconstructed trajectories captures those obtained directly from MD trajectories. They also use obtained GNN to provide mechanistic insights about allosteric regulations in these three systems. This is a novel approach and applications of three different systems also show the strength of the approach. However, the method needs more testing and detailed comparison with the other MD based methods in the field before it gets published. Only then the main and the most important conclusion, that is their NRI based model can capture the interaction patterns which cannot be obtained by other MD method, can be justified. The manuscript should be revised by addressing the issues discussed below

Response: We thank the reviewer for the enthusiastic comments on our approach and also greatly appreciate all the constructive suggestions and critiques, which have helped us further improve the quality of our manuscript. Our responses are as follows.

- (i) It is not clear how their GNN approach is better in capturing non-linear correlation that cannot be capture with co-variances matrices or cross-correlations maps of MD analysis. The findings obtained from GNN should compared with other MD based approaches to show that it is indeed novel, and capture non-linear correlations as stated.(See related allostery works by HX Zhou, Barahona, McCullagh, Ozkan, Gohlke)

Response: We agree that more work is needed to justify the novelty of our approach. As suggested, we surveyed the works from PubMed and showcased the comparisons of our NRI-based analysis with three positional covariance-based approaches (constraint network analysis, derivative centrality metric of the Hessian, and dynamics coupling index).

First, we performed MD simulations and NRI model training for the wild type (WT) and 23 Ala-mutants of human Pin1. Given the graph learned from the NRI model, we calculated the difference of the learned edges (pairwise free energy, $G_{Z_{ij}}$) between the WT and 23 Ala-mutants to reflect the change in relative free energy caused by Ala-mutation, $\Delta G_Z = E_Z^{Ala-mutant} - E_Z^{WT}$. We compared the difference of unfolding free energies ($\Delta\Delta G$) between the wild type and mutants from the chemical denaturation experiments (**Fig. 6a and b**), with ΔG_Z . The correlation between the computed relative free energy (ΔG_Z) based on our model and the experimental free energy ($\Delta\Delta G$) is significant ($R^2 \geq 0.931, p \leq 1.166 \times 10^{-10}$) (**Fig. 6c and d**). In contrast, the comparison between free energy (ΔG_{CNA}) computed from the constraint network analysis proposed by *Holger Gohlke's* group and experimental free energy shows a relatively poor correlation ($R^2 = 0.188, p = 0.390$) (**Fig. 6e**). Almost no correlation ($R^2 = -0.093, p = 0.671$) is observed between the potential energy (ΔG_{Total}) from MD simulations and the experimental data (**Fig. 6f**). Our model has dramatically improved the free energy estimate compared with the CNA approach and MD-based method.

Fig. 6: Evaluations of energy calculation performance of the NRI-based approach. **a** shows a summary of thermodynamic data for the WT and 23 Ala-mutants of Pin1. N.D., not determined, represents that the mutant is too unstable to measure. **b** shows the effects of the Ala mutations on the equilibrium stability of Pin1. A positive value of ($\Delta\Delta G = \Delta G^{Ala-mutant} - \Delta G^{WT}$) indicates that the Ala mutation is destabilizing relative to the wild type. Mutations that destabilize more than 3 kcal/mol ($\Delta\Delta G \geq 3$ kcal/mol) are shown as red bars, more than 1 kcal/mol, and less than 3 kcal/mol ($1 \leq \Delta\Delta G \leq 3$ kcal/mol) are shown as blue bars. **c** and **d** show a correlation between the computed relative free energy ($\Delta G_Z = E_Z^{Ala-mutant} - E_Z^{WT}$) and $\Delta\Delta G$ from the chemical denaturation experiments for the 23 Ala-mutants of Pin1. Based on the position vector of C α in Pin1 (PDB ID: 1NMV), the threshold of residue-residue distance was set to 12 Å and 15 Å to present the residue-level interaction. **e** shows the correlation between the predicted ΔG_{CNA} (Eq. (3) in Supplementary Method 3) and $\Delta\Delta G$ for the 23 Ala-mutants of Pin1. **f** shows the correlation between the $\Delta G_{Total} = G_{Total}^{Ala-mutant} - G_{Total}^{WT}$ obtained from MD simulations and $\Delta\Delta G$ for the 23 Ala-mutants of Pin1.

Second, we validated our NRI model on the Pin1 case study: positive and negative regulation in FFpSPR- and pCdc25C- bound Pin1. The stability maps and per-residue free energy obtained by using the CNA approach show that only residues in the WW domain can be identified as impactful on the structural stability upon the ligand binding (**Supplementary Fig. 21a**). In comparison, the interaction maps learned by the NRI model present sufficient information related to allosteric communication (**Supplementary Fig. 21b**).

Third, we showcased the comparison of the results learned from our model and the calculation of the derivative centrality metric proposed by *Martin McCullagh*'s group. As presented in **Fig. 7a**, the complete allostery propagation was detected after 200 ns (frame 2000) by measuring the derivative centrality metric. In comparison, the NRI-based approach can capture the allosteric signal propagated to the catalytic sites before observing the complete open-to-closed transition (simulation time = 108 ns) (**Fig. 7c** and **Supplementary Fig. 4a**).

Finally, we showcased the comparison of the results learned from our model and the calculation of the dynamic coupling index (DCI) proposed by the groups of *Huan-Xiang Zhou* and *S. Banu Ozkan* (**Supplementary Fig. 25**). The results show that our model is sensitive enough to detect the signal conveying to the catalytic loop in the first 50 ns (frames 1-500) as indicated in the more robust weights in the catalytic loop than other domains in Pin1. More details of the results have been added in the **Results**

section (Pages 14-18): Performance comparison between the NRI model and three covariance-based models.

a FFpSPR-Pin1

b pCdc25C-Pin1

c FFpSPR-Pin1

d pCdc25C-Pin1

Fig. 7: Comparison of the performance of the Hessian-based and NRI-based approaches in capturing the allosteric signals in simulations. a and b present the derivative node metrics δ_{node} based on Hessian as a function of residues for FFpSPR- and pCdc25C-bound Pin1 systems. The metrics are calculated by using frames 1-500, 1-1000, ..., 1-5000 of the trajectories. Residues that have strong connections are highlighted by vertical bars, whose colors depend on the domains. **c and d** present the distribution of learned edges between the domains and the corresponding average conformation mapped

with RMSD value for FFpSPR- and pCdc25C-bound Pin1 system. The edges are learned from frames 1-500, 1-1000, ..., 1-5000 of the trajectories.

- (ii) There is also lack of discussion/comparison that are specific to these three systems. All of the three systems have been extensively studied by other computational groups. The authors should add a discussion whether their results overlap with those findings, or whether their results differ and possible discussion about the sources of difference and overlaps.

Response: It is a good point. As suggested, we added the discussion and comparison of our findings and the results studied by other computational groups. We have some findings consistent with some recent results. For example, the MD simulation studies by *Huan-Xiang Zhou's* group noted that the presence of the positive allosteric ligand FFpSPR enhances the interdomain interaction between the WW domain and the PPIase domain through two pathways. Our results not only show the strengthened interdomain contact but also identified another pathway of open communication from the WW through the $\alpha 1$ and $\alpha 2-3$ helices, ending in the catalytic loop (see Fig. 3a, *right*). In addition, we investigated the effects of the positive and negative ligands on the conformational transition of the well-separate structure, which agrees with the negative mechanism. Please check the **Discussion section on Page 19** for details.

- (iii) In same context, the identified results are not compared or discussed with the available experimental data. For example, NMR analysis of PIN1-WW suggest a negatively controlled allosteric regulation where substrate binding of WW domain a reduces the interdomain contact as compared to that of the Apo state, thus allowing the PPIase domain to freely search for a distinct pS/T-P substrate. This plausible mechanism suggested by the results of NMR analysis does not completely agree with the findings of GNN analysis of the present work. Why is that ? This brings the issue of the approach and justification of simulation and sampling.

Response: We would like to thank the reviewer for the suggestion. We have added the study of negative regulation in the **Results section of the first case study on Page 8: Pathways mediate inter-domain allosteric communication in Pin1**. We used Pin1 with two domains (the WW domain and the PPIase core) well-separated as our starting structure to perform simulations of the FFpSPR- /pCdc25C- bound forms and trained the corresponding trajectories using the NRI model. The distribution of learned edges of pCdc25c-Pin1 shows that only the PPIase core interacts with the WW domain. Almost no edges connect to the catalytic loop, reflecting the reduced intradomain contacts in the PPIase domain (**Figs. 2b-d and 3e**). The ligand FFpSPR enables the interaction with the catalytic domain by enhancing the communication from the WW domain through the PPIase core ending to the catalytic loop (**Figs. 2b-d and 3d, Supplementary Fig. 3**).

- (iv) In their approach, the training of NRI (their learning system) depends on sampling of the trajectories (basically output frequency). There is no justification of choosing 50 frames. They should add a section where they study impact of sampling frequency on the edges learned and their weight. What is the uncertainty on those edges and their weights ?One would expect to see variation in long distance coupling with lower sampling frequency.

Response: As suggested, we added a section in the manuscript to evaluate the effect of sampling frequency on the NRI model performance. We ran the model for three case studies with 10, 15, 20, 25, 30, 40, 50, 60, 75, 90, and 100 steps. Results for the MSE and VSD values are shown in **Supplementary Fig. 14**. Then, we clustered the MD trajectories using the K-means clustering algorithm (**Supplementary Fig. 15**). Finally, we showcased the distribution of learned edges for three case studies (**Supplementary Figs. 16-19**). Across all results with different sampling steps, we found that (i) the low-frequency sampling results in fewer edges weights because of feeding in less structural information; (ii) since the conformational samples obtained from MD simulations are diverse and abundant, using small frames to represent the complete conformational states may be too coarse-grained and ignore too many details. Hence, we selected the learning results based on both the small reconstruction error and sufficient sampling. Please check the

Results section on Pages 13-14: Effects of sampling frequency on the learned edges and their weights for details.

- (v) This as brings up another point of simulation time. The authors used 200 ns simulations after the equilibration. I recommend increasing the simulation time to 500 to 1 micro seconds, and will the results obtained from longer trajectories still capture the same long-range coupling and pathways emerged from the 200ns? Will the weights of edges, connecting different domains remain same ? This analysis is necessary to show the power and accuracy of the approach.

Response: We increased the simulation time to 500 ns for Pin1 and SOD1 systems, and increased the simulation time to 1 μ s for the MEK1 system. At the same time, we repeated the simulations twice for three cases to validate the power and accuracy of our approach. The results of learned edges are shown in **Supplementary Fig. 20**, which demonstrates that the edges learned for three repeat trajectories remain almost the same. We also added this result in the **Results section (Pages 13-14): Effects of sampling frequency on the learned edges and their weights**.

Reviewer #2 (Remarks to the Author):

Key Results:

This paper applies the neural relational inference (NRI) model to learn the latent interactions among residues of proteins given the molecular dynamics (MD) trajectories. The idea is to treat the protein as a graph where each node is a residue, each edge represents the interaction between two residues, and the node feature includes the 3D velocity and 3D position. The NRI model is trained to reconstruct the given MD trajectories of proteins.

Significance:

I am incompetent in judging the significance of this paper to the related biological fields. From the perspective of machine learning (ML), this paper basically applies the neural relational inference model proposed in [1] to learn the latent interaction among residues of proteins. It is a novel and useful application of this ML model.

[1] Kipf, T., Fetaya, E., Wang, K.C., Welling, M. and Zemel, R., 2018, July. Neural relational inference for interacting systems. In International Conference on Machine Learning (pp. 2688-2697). PMLR.

Response: We thank the reviewer for the positive comments and also greatly appreciate the following constructive suggestions. Please find our point-by-point responses as follows.

Data and Methodology:

The data and non-machine-learning part of the methodology are outside the scope of my expertise. From the perspective of machine learning, the design of applying the NRI model to solve the problem is technically sound, i.e., treating the protein as a graph and use NRI to learn the latent interactions among residues as well as reconstructing molecular dynamics trajectories. Once trained, the decoder along with the fixed prior of the NRI constructs a generative model. I am curious how good the generated trajectories are from this generative model. Are there any useful applications for this generative model in the current problem context?

Response: To evaluate the accuracy of the generated trajectories, we calculated the value square deviation (VSD) of RMSD value between the ground truth and reconstructed trajectories. The results are shown in **Supplementary Fig. 2** for the Pin1 study, **Supplementary Fig. 9** for the SOD1 study, and **Supplementary Fig. 12** for the MEK1 study. The VSD values, on average, are less than 0.2 for all three systems, which shows that the conformation distribution of the generated trajectories is almost the same as that of the true trajectories. Although this does not suggest the exact match of atomic positions at each time point, which is not meaningful in MD simulations anyway, it indicates that the statistical properties are probably the same between the ground truth and reconstructed trajectories, which is sufficient for essentially all MD simulation purposes.

In the revised manuscript, we concluded two useful applications for this generative model.

(i) This model has the power to calculate the difference of relative free energy $\Delta G_Z = E_Z^{Ala-mutant} - E_Z^{WT}$ caused by residue mutation. The results demonstrate that the relative free energy computed using the NRI-based approach agrees very well with the experimental data. The details are shown in **Fig. 6** and the Result section (**Pages 14-15**): **Performance comparison between the NRI model and three covariance-based models.**

(ii) Since the NRI model can accurately reflect the ground-truth trajectories, it can present the interaction patterns and signal-transmit pathways for protein biological functions. Hence, such a study can be used as a guide for protein mutation design or drug design, which is vital for disease studies. We also added some discussions on **Page 19**.

Validity & Analytical Approach:

From the perspective of machine learning, I have two concerns regarding the evaluation.

- (1) It is mentioned that traditional ML methods like principal component analysis (PCA) and cross-correlation analysis (CCA) were applied to this problem in the literature. They are restrictive due to their linearity. However, I did not find any comparison of NRI versus these traditional methods in any form. It would be necessary to include such comparisons since one of the main motivations is that NRI can model nonlinear correlation in dynamics whereas traditional methods can not.

Response: It is a good point. Recently, many positional covariance-based approaches were applied in allosteric studies. We selected three methods (constraint network analysis, derivative centrality metric of the Hessian, and dynamics coupling index) to compare their performance with our NRI-based approach. Since principal component analysis is rarely used in the study of long-term regulation, this study did not compare the principal component analysis with our method.

First, we performed MD simulations and NRI model training for the wild type (WT) and 23 Ala-mutants of human Pin1. Given the graph learned from the NRI model, we calculated the difference of the learned edges (pairwise free energy, $G_{Z_{ij}}$) between the WT and 23 Ala-mutants to reflect the change in potential energy caused by Ala-mutation, $\Delta G_Z = E_Z^{Ala-mutant} - E_Z^{WT}$. We compared the difference of unfolding free energies ($\Delta\Delta G$) between the wild type and mutants from the chemical denaturation experiments (**Fig. 6a and b**), with ΔG_Z . The correlation between the computed relative free energy (ΔG_Z) based on our model and the experimental free energy ($\Delta\Delta G$) is significant ($R^2 \geq 0.931, p \leq 1.166 \times 10^{-10}$) (**Fig. 6c and d**). In contrast, the comparison between free energy (ΔG_{CNA}) computed from the constraint network analysis proposed by *Holger Gohlke* and experimental free energy shows a relatively poor correlation ($R^2 = 0.188, p = 0.390$) (**Fig. 6e**). Almost no correlation ($R^2 = -0.093, p = 0.671$) is observed between the potential energy (ΔG_{Total}) from MD simulations and the experimental data (**Fig. 6f**). Our model has dramatically improved the free energy estimate compared with the CNA approach and MD-based method.

Second, we validated our NRI-based model on the Pin1 case study: positive and negative regulation in FFpSPR- and pCdc25C- bound Pin1. The stability maps and per-residue free energy obtained by using the CNA approach show that only residues in the WW domain can be identified as impactful on the structural stability upon the ligand binding (**Supplementary Fig. 21a**). In comparison, the interaction maps learned by the NRI-based model present sufficient information related to allosteric communication (**Supplementary Fig. 21b**).

Third, we showcased the comparison of the results learned from our model and the calculation of the derivative centrality metric proposed by *Martin McCullagh's* group. As presented in **Fig. 7a**, the complete allostery propagation was detected after 200 ns (frame 2000) by measuring the derivative centrality metric. In comparison, the NRI-based approach can capture the allosteric signal propagated to the catalytic sites before observing the complete open-to-closed transition (simulation time = 108 ns) (**Fig. 7c and Supplementary Fig. 4a**).

Finally, we showcased the comparison of the results learned from our model and the calculation of the dynamic coupling index (DCI) proposed by the groups of *Huan-Xiang Zhou* and *S. Banu Ozkan* (**Supplementary Fig. 25**). The results show that our model is sensitive enough to detect the signal conveying to the catalytic loop in the first 50 ns (frames 1-500) as indicated in the more robust weights in the catalytic loop than other domains in Pin1. More details of the results have been added in the **Results section (Pages 14-18): Performance comparison between the NRI model and three covariance-based models**.

- (2) Since the main goal of the proposed model is to learn the latent interaction of residues, it is necessary to quantitatively measure how accurate these learned latent interactions are. I am not sure whether authors could obtain the ground-truth values of these latent variables for a subset of data.

If that is not the case, it would still be necessary to obtain some indirect quantitative metrics. I did not find any quantitative metrics in the current draft.

Response: We would like to thank the reviewer for the suggestion. The question is partially answered in the comment above. To quantitatively evaluate the performance of our model, we performed MD simulations and NRI model training for the wild type (WT) and 23 Ala-mutants of human Pin1. Given the graph learned from the NRI model, we calculated the difference of the learned edges (pairwise free energy, G_{Zij}) between the WT and 23 Ala-mutants to reflect the change in potential energy caused by Ala-mutation, $\Delta G_Z = E_Z^{Ala-mutant} - E_Z^{WT}$. We compared the difference of unfolding free energies ($\Delta\Delta G$) between the wild type and mutants from the chemical denaturation experiments (Fig. 6a and b), with ΔG_Z . The correlation between the computed relative free energy (ΔG_Z) based on our model and the experimental free energy ($\Delta\Delta G$) is significant ($R^2 \geq 0.931, p \leq 1.166 \times 10^{-10}$) (Fig. 6c and d). We believe this is a convincing quantitative metric to evaluate the power and accuracy of the NRI model. More details of results have been added in the **Results section (Pages 14-15): Performance comparison between the NRI model and three covariance-based models.**

Suggested Improvements:

Please see the other comments for specific suggestions.

Clarity and Context:

- (1) The decoder part is not clearly explained. In particular, from Eq. (5), it seems that the autoregressive conditional probability on the right-hand side of Eq. (3) is a Gaussian. However, it is not explicitly mentioned anywhere what kind of distribution is being used. From line 403 to line 413, it is unclear how the recurrent update and the GNN are mixed to implement this autoregressive conditional probability. Furthermore, Eq.(5) is not accurate in describing the dependency on the latent variable z since z does not appear on the right-hand side. It would be better to clearly explain how μ and σ depend on z .
- (2) Some mathematical notations need to be improved and/or clarified. For example, the subscript of the encoder used in Eq. (1) is very confusing and not explained anywhere.
- (3) It would be great to clearly explain what the $K=4$ edge types are in line 394.
- (4) What is the exact performance metric used during the validation to determine the best hyperparameters?

Response: We apologize for the unclear description and ambiguous equations, which led to the confusion. We double-checked the method and changed the order of some equations (see Pages 21-23). The revised version is shown as follows.

- (1) The reconstruction error $\mathbb{E}_{q_{\Phi(z|x)}}[\log p_{\theta}(x|z)]$ assumes the predicted outputs represent means of a Gaussian distribution. In the equation $\mathbb{E}_{q_{\Phi(z|x)}}[\log p_{\theta}(x|z)] = -\sum_j \sum_{t=2}^T \frac{\|x_j^t - \mu_j^t\|^2}{2\sigma^2} + const$, σ^2 is a fixed variance, x_j^t is the correct input, and μ_j^t is the predicted mean computed in the decoder operation. We added the encoder and decoder operations in the method as follows:

More formally, as shown in Fig. 1d (on the left), the encoder takes the form

$$q_{\Phi}(z_{ij}|x) = \text{softmax}(f_{enc,\Phi}(x)_{ij,1:K}) \quad (2)$$

in which $f_{enc,\Phi}(x)$ is a GNN performed on the fully connected networks (without self-connection) to predict the latent graph structure. The encoder operation is presented as follows:

$$h_j^l = f_{emb}(x_j) \quad (3)$$

$$h_{(i,j)}^l = f_e^l([h_i^l, h_j^l]) \quad (4)$$

$$h_j^{l+1} = f_v^l(\sum_{i \neq j} h_{(i,j)}^l) \quad (5)$$

$$h_{(i,j)}^{l+1} = f_e^{l+1}([h_i^{l+1}, h_j^{l+1}]) \quad (6)$$

where h_j^1 is the embedding of node v_j in layer l , $h_{(i,j)}^l$ is an embedding of the edge $e_{(i,j)}$. Equations (4)-(5) represent node-to-edge ($v \rightarrow e$) and edge-to-node ($e \rightarrow v$) operations, respectively.

The decoder is expressed as:

$$p_\theta(x|z) = \prod_{t=1}^T p_\theta(x^{t+1}|x^t, \dots, x^1, z) \quad (8)$$

which reconstructs the dynamic systems $p_\theta(x^{t+1}|x^t, \dots, x^1, z)$ with a GNN given the latent graph structure z . A recurrent decoder with a GRU unit is required to model $p_\theta(x^{t+1}|x^t, \dots, x^1, z)$. The decoder operation is presented as follows:

$$\tilde{h}_{(i,j)}^t = \sum_k z_{ij,k} \tilde{f}_e^k([\tilde{h}_i^t, \tilde{h}_j^t]) \quad (9)$$

$$MSG_j^t = \sum_{i \neq j} \tilde{h}_{(i,j)}^t \quad (10)$$

$$\tilde{h}_j^{t+1} = GRU([MSG_j^t, x_j^t], \tilde{h}_j^t) \quad (11)$$

$$\mu_j^{t+1} = x_j^t + f_{out}(\tilde{h}_j^{t+1}) \quad (12)$$

$$p(x^{t+1}|x^t, z) = \mathcal{N}(\mu^{t+1}, \sigma^2 \mathbf{I}) \quad (13)$$

in which $z_{ij,k}$ is the k -th element of the vector z_{ij} , σ^2 is a fixed variance, x_j^t is the correct input, μ_j^t is the predicted mean, and f_{out} denotes an output transformation.

(2) We added the explanations for mathematical notations.

This model is formalized as a variational autoencoder (VAE) that maximizes the evidence lower bound (ELBO):

$$\mathcal{L}(\Phi, \theta) = \mathbb{E}_{q_\Phi(z|x)} [\log p_\theta(x|z)] - KL[q_\Phi(z|x) || p_\theta(z)] \quad (1)$$

in which Φ and θ are trainable parameters of probability distributions.

(3) We added the explanations for $K=4$ edge types.

The interactions between nodes i and j take the form of a latent variable $z_{i,j} \in \{1, \dots, K\}$, in which K is the number of interaction types being modeled. These interaction types do not have any pre-defined meaning, but rather the model learns to assign a meaning to each type. In our modeling, for easy visualization, we used $K=4$ types to train the model. The first edge type is “hard-coded” as non-edge and was trained with a prior probability of 0.91. All other edge types received a prior of 0.03 to favor sparse graphs.

(4) We mainly used MSE (Mean Squared Error) to evaluate the performance of trajectory prediction in the hyperparameter tuning. Please check **the Results section on Pages 13-14: Effects of sampling frequency on the learned edges and their weights**.

References:

It would be great to cite the original GNN paper [2] since GNN has been mentioned many times throughout the paper.

[2] Scarselli, F., Gori, M., Tsoi, A.C., Hagenbuchner, M. and Monfardini, G., 2008. The graph neural network model. IEEE transactions on neural networks, 20(1), pp.61-80.

Response: As suggested, we cited this GNN paper in the manuscript.

Reviewers' Comments:

Reviewer #1:

Remarks to the Author:

The authors have worked on in details to answer concern raised, which were not clear in the first version. However, I still think this revised version has some points that are needed to be addressed.

(i) The authors claims that their approach capture negatively controlled allosteric regulation by pCdc25c, presenting distribution of learned edges of pCdc25c- Pin1 where only the PPIase core interacts with the WW domain. However, to obtain these results, they started a "hypothetical open conformation" where the linker region connecting the two domain is modified. I There is no reasoning why such a different conformationally never observed (through NMR studies) is chose. It looks like a tailoring a start conformation to reduce the interaction or learned edges between the catalytic loop and WW domain.

(ii) Second, as suggested, the authors have increased the simulations times to 500 ns for Pin1 and SOD1 systems, and to 1 μ s for the MEK1 system. Although, they stated that the learned edges and weights remain the same, I see significant differences, particularly the repeats are compared for WW-FFpSPR (1NMV), AV52Mek1 they did not address this.

(iii) The effect of sampling frequency on NRI really alters distribution of the learn edges, besides the weights keep changing based on sampling frequency, and some edges appear then disappear even at higher sampling frequency , which makes choice of sampling frequency (50) arbitrary. Moreover, the constant increase VSD values shows that the approach fails to capture long time scale behaviors (which underlies indeed protein function).

(iv) I also notice some problems about comparison with other co-variance based approaches that need to be addressed. For example, to compare the energies obtained pairwise interaction of learned edges versus folding stability of alanine mutation of 23 positions, the bound simulation model is used. That does not make sense, instead Apo (unbound model) should be used if they want to compare the folding stability. As another note, the computed pairwise energies are not "free energies" they are completely empirical so they should not call them free energies.

They also need to show how their approach give such a strong (but do not make sense as they use an bound state) correlation versus other method did not. They should detail which interactions were missed in the other method.

(v) As a minor point, claiming determining the long-range communication with their approach in 50 ns versus some other approach 100 ns does not show an improvement, in MD simulations scale, particularly with GPU processes the time to simulate 50 ns vs 100 ns differ only couple hours and then I think NRI models takes more time to compute than the other methods compared.

Reviewer #2:

Remarks to the Author:

I would like to thank the authors for providing a thorough response and careful revise of the draft. The responses from the authors resolve most of my concerns, especially adding a new application for the generator, i.e., evaluation of the difference of relative free energy. Newly added experiments show that the proposed NRI model outperforms the traditional positional covariance-based approaches in terms of the proposed metrics. The clarification on the model part also helps improve the readability of the paper significantly.

From the machine learning perspective, the only concern I have now is about the importance of learning the interaction versus not for NRI in molecular dynamics (MD) simulations. In particular, it would be great to add one VAE baseline which does not have latent variables over edges. Instead, its latent variable per time step would be a Gaussian vector (the latent dimension is a hyperparameter) per node. This can be easily achieved by replacing the softmax readout (over the edge) of the current encoder as Gaussian readout (over the node) like what the original VAE does. The prior would

accordingly become the independent standard Gaussian per node. The GNN based encoder and decoder will be both operated on fully connected graphs. The decoder readout layer would still be Gaussian per node. By doing so, we remove the latent interaction graph and instead use fully connected one. Therefore, it could help understand how important it is to learn the interaction for the purpose of generating faithful MD trajectories.

I agree that having a NRI to learn the interactions would greatly improve the interpretability. But it is unclear whether it is necessary for the MD simulation task. It could be the case that without explicit modeling of interactions, VAEs could get better MD simulations. Therefore, comparing with such a baseline would be more convincing.

REVIEWER COMMENTS

Reviewer #1 (Remarks to the Author):

The authors have worked on in details to answer concern raised, which were not clear in the first version. However, I still think this revised version has some points that are needed to be addressed.

Response: We thank the reviewer for the positive comments on our approach and also greatly appreciate all the constructive suggestions and critiques, which have helped us further improve the quality of our manuscript. Our responses are as follows.

- (i) The authors claims that their approach capture negatively controlled allosteric regulation by pCdc25c, presenting distribution of learned edges of pCdc25c- Pin1 where only the PPIase core interacts with the WW domain. However, to obtain these results, they started a “hypothetical open conformation” where the linker region connecting the two domain is modified. I There is no reasoning why such a different conformationally never observed (through NMR studies) is chose. It looks like a tailoring a start conformation to reduce the interaction or learned edges between the catalytic loop and WW domain.

Response: We are sorry that we did not make this clear. In fact, the “open conformation” (PDB ID: 1NMV) used in the study was observed through NMR study by *Elena Bayer et al. (Bayer, E., Goettsch, S., Mueller, J. W., Griewel, B., Guiberman, E., Mayr, L. M., & Bayer, P. (2003). Structural analysis of the mitotic regulator hPin1 in solution: insights into domain architecture and substrate binding. Journal of Biological Chemistry, 278(28), 26183-26193.)* We cited this paper in the revised version (ref. 24). This paper demonstrated that the extended conformation is shifted towards its closed conformation upon substrate binding to the WW domain of Pin1 structure. Furthermore, we also added the simulation of “closed conformation complexed with pCdc25C” to probe the negative regulation. The learned edges between domains are illustrated in Figure 2. We summarized that the pCdc25C binding to the WW domain of the closed Pin1 reduces the interactions between the WW domain and the PPIase core. Besides this observation, the edges in the distribution interaction graph of the PPIase domain are reduced compared to the positive FFpSPR binding to the WW domain (Figure 2b-c, iv). We clarified this in the **Results section (Page 9)** under “Pathways mediate inter-domain allosteric communication in Pin1”:

The strengthened Pin1 interdomain contact upon the FFpSPR binding is referred to as positive regulation. In addition, an NMR study²⁴ of Pin1-WW suggests a negative regulation, i.e., a negative allosteric peptide pCdc25C (EQPLpTPVTDL) binding to the WW domain reduces the interdomain contact, thus allowing the PPIase domain to search for a distinct pS/T-P substrate freely. To further investigate the effects of the pCdc25C binding on interdomain contact, we simulated the pCdc25C-Pin1 complex (PDB ID: 1PIN). We found that the pCdc25C binding to WW domain of the closed Pin1 reduces the interactions between the WW domain and the PPIase

core. Besides this, the edges in the networks of PPIase domain are reduced compared to the positive FFpSPR binding to the WW domain (Fig. 2b-d, iv). In addition, we used Pin1 with two domains (the WW domain and the PPIase core) well separated as our starting structure (PDB ID: 1NMV) to perform the simulations of the FFpSPR- /pCdc25C- bound forms and trained the corresponding trajectories using the NRI model. From the representative conformations clustered from the trajectories (Supplementary Fig. 4), we observed that the positive allosteric ligand FFpSPR promotes an open-to-closed transition within 108 ns. However, the peptide pCdc25C binding produces a range of diverse and separate conformations. The distribution of learned edges of pCdc25c-bound Pin1 shows that only the PPIase core interacts with the WW domain. Almost no edges connect to the catalytic loop, reflecting the reduced intradomain contacts in the PPIase domain (Figs. 2b-d, vi, and 3e). The ligand FFpSPR enables the interaction with the catalytic domain by enhancing the communication from the WW domain through the PPIase core ending to the catalytic loop (Figs. 2b-d, v, and 3d, Supplementary Fig. 3).

- (ii) Second, as suggested, the authors have increased the simulations times to 500 ns for Pin1 and SOD1 systems, and to 1 μ s for the MEK1 system. Although, they stated that the learned edges and weights remain the same, I see significant differences, particularly the repeats are compared for WW-FFpSPR (1NMV), AV52Mek1 they did not address this.

Response: We agree that the statement of "exactly the same" is not accurate. Nevertheless, despite some differences in the learned edges, the topologies of underlying pathways that determine signaling are similar for repeated trajectories. As shown in the figure below, we present the topologies of interactions with the weights of the edges greater than 0.7. The results from trajectories 1 and 3 show that the WW domain acts directly with the PPIase core during the simulation. For trajectory 2, the WW domain has the possibility to interact with both PPIase core and helices, but the WW domain is more inclined to interact with PPIase core in terms of more intensive weight values. Thus, the topology between WW domain and PPIase core is stable in these repeated learning sets. The calculated allosteric pathways of FFpSPR-Pin1 (PDB ID: 1NMV) (Supplementary Fig. 29) shows that even though the weights of the three learned edges are different, the shortest pathways from the WW domain to the catalytic loop are similar, as they all start from the WW domain and pass through the PPIase core before passing to the catalytic loop. Moreover, the topologies from WW domain to helices and from WW domain to PPIase core are stable in the underlying pathways obtained from the repeats of FFpSPR-Pin1 (PDB ID: 3TDB) (Supplementary Fig. 29). For the SOD1 system (Supplementary Fig. 30), the NRI model is also able to capture highly consistent topologies of underlying pathways, i.e., the small active loops (DL, DiL, and ZL) stabilize the closure of the electrostatic loop. For the MEK1 system (Supplementary Fig. 31), the difference in the edges in MEK1 system is slightly larger. However, the important topological elements (activation segment and proline-rich loop) are learned to illustrate the signal transmitting.

FFpSPR-Pin1 (PDB ID: 1NMV)

Edge weight > 0.7

Response Figure 1: The distribution of learned edges for the three repeated trajectories of FFpSPR-Pin1 (PDB ID: 1NMV). The edge weights shown here are greater than 0.7.

To illustrate this in depth, we changed some wording and added new text in the **Results section (Pages 14-15)** under “Effects of sampling frequency on the learned edges and their weights”:

In our study, the simulations for three case studies were repeated two additional times to validate the power and accuracy of our approach. The edges learned for the three repeated trajectories remain similar but have some differences, especially in the Pin1 and MEK1 case studies (Supplementary Fig. 28). Thus, we calculated the network node centralities (representing the importance of a residue) in allosteric pathways for the three case studies (Supplementary Figs. 29-31) and observed that the residues in the PPIase core play a crucial allosteric signal transmitting role in all three dynamic regulations of FFpSPR-Pin1 (PDB ID: 1NMV). Upon the FFpSPR binding to the WW domain of the extended Pin1 structure, the interactions between the WW domain and the catalytic loop are supported by the edges directly connected from the WW domain to the PPIase core. Thus, the topology between WW domain and PPIase core is stable in these repeated learning sets. Moreover, the topologies from the WW domain to helices and from the WW domain to the PPIase core are also stable in the underlying pathways obtained from the repeats of FFpSPR-Pin1 (PDB ID: 3TDB) (Supplementary Fig. 29).

For the SOD1 system (Supplementary Fig. 30), the NRI model is also able to capture highly consistent topologies of the underlying pathways; in particular, the small active loops (DL, DiL, and ZL) stabilize the closure of the electrostatic loop. As for the A52V MEK1 study, the repeats demonstrated that the networks in the WT and A52V MEK1s are sparse compared with the

MEK1 upon the active mutated. For both SOD1 and Pin1 systems, the allosteric pathways are almost learned reproducibly with fewer differences than MEK1. The difference in the edges in MEK1 system is slightly larger. Nevertheless, the important topological elements (activation segment and proline-rich loop) are learned to illustrate the signal transmitting. Due to the chaotic and stochastic nature of molecular dynamics simulations, identical trajectories cannot be obtained even with the same set of parameters. However, the NRI model is still able to extract the key allosteric pathways related to protein dynamics regulation consistently, suggesting the model is robust.

Supplementary Figure 29: The node centralities in the allosteric pathways between the WW domain and the catalytic loop for three repeated trajectories of FFpSPR-Pin1 complexes. The color scale bar (from 1.0->0.0) represents the decrease in the importance of a residue measured by normalized node centrality (i.e., the fraction of suboptimal paths going through the node or residue).

Supplementary Figure 30: The node centralities in the allosteric pathways between residue G93A and the electrostatic loop for three repeated trajectories of WT- and G93A-SOD1 complexes.

Supplementary Figure 31: The node centralities in the allosteric pathways between the activation segment and the α C-helix/proline-rich loop for three repeated trajectories of S218Sp/S222Sp- and E203K-MEK1 complexes.

(iii) The effect of sampling frequency on NRI really alters distribution of the learn edges, besides the weights keep changing based on sampling frequency, and some edges appear then disappear even at higher sampling frequency, which makes choice of sampling frequency (50) arbitrary. Moreover, the constant increase VSD values shows that the approach fails to capture long time scale behaviors (which underlies indeed protein function).

Response: Thank you for pointing out this important issue. Like most other methods, it requires some reasonable ranges for the parameters in order to make NRI work. To compare the effect of different sampling steps on the results consistently, we first unified the total length of the training trajectories. Different frequencies of sampling are achieved by varying the interval of each step. We have demonstrated that with suitable sampling frequencies (50 to 100 steps) and a suitable step interval (~20 ns), the NRI model can reveal long-range allosteric interactions in slow-motion of up to 1 μ s. Furthermore, we did further analysis during this revision to demonstrate that the method is robust over the selection of sampling frequency. The results have been added in the **Results section (Pages 13-14)** under “Effects of sampling frequency on the learned edges and their weights”:

To study the effects of step intervals in the NRI model learning, we learned the trajectories with different time intervals of sampling. Furthermore, we compared the RMSF values between the actual and the reconstructed trajectories, where a lower VSD value means a better model (Supplementary Figs. 20, 22, 24, and 26). It is observed that the reconstruction error slightly increases as the sampling step interval decreases when the total duration of the training trajectory keeps the same, possibly because it is harder to reconstruct trajectory details as the step interval decreases. Nevertheless, the reconstructed trajectory matches the actual trajectory relatively well even when the step interval is reduced to 5ns or 4ns (sampling frequency increase to 100 and 50 steps) for 500 ns’ SOD1 simulation and 200 ns’ Pin1 simulations (Supplementary Figs. 20 and 26). It is worth noting what step interval to use may depend on biological systems. For example, a sampling step much longer than 20 ns may be too long to recover enough information in the allosteric process (Supplementary Figs. 21, 23, 25, and 27). Our results show that a step interval of ~20 ns can yield a more reasonable outcome.

Supplementary Figure 20: Comparison of RMSF values between the simulation and the reconstruction of trajectories for WT-SOD1 (a) and G93A-SOD1 (b). The trajectories of WT and G93A-SOD1 were modeled using time intervals of 50, 20, 10, and 5 ns (sampling steps of 10, 25, 50, and 100, respectively). The total simulation time is 500 ns.

Supplementary Figure 21: The distribution of learned edges for WT-SOD1 (a) and G93A-SOD1 (b), obtained from the modeling using time intervals of 50, 20, 10, and 5 ns (sampling step with 10, 25, 50, and 100, respectively). The total simulation time is 500 ns.

Supplementary Figure 22: Comparison of RMSF values between the simulation and the reconstruction of trajectories for apo Pin1 (a), FFpSPR-bound Pin1 (b), FFpSPR-bound Pin1 (I28A) (c). The trajectories of the Pin1 complexes were modeled using time intervals of 50, 20, and 10 ns (sampling step with 10, 25, and 50, respectively). The total simulation time is 500 ns.

Supplementary Figure 23: The distribution of learned edges for the Apo-Pin1 (a), FFpSPR-bound Pin1 (b), and FFpSPR-bound Pin1 (I28A) (c), obtained from the modeling using time intervals of 50, 20, and 10 ns (sampling step with 10, 25, and 50, respectively). The total simulation time is 500 ns.

Supplementary Figure 24: Comparison of RMSF values between the simulation and the reconstruction of trajectories for FFpSPR-bound Pin1 with two domains-separated (**a**), and pCdc25C-bound Pin1 with two domains-separated (**b**). The trajectories of Pin1 complexes were modeled using time intervals of 50, 20, and 10 ns (sampling step with 10, 25, and 50, respectively). The total simulation time is 500 ns.

Supplementary Figure 25: The distribution of learned edges for FFpSPR-bound Pin1 with two domains-separated (a), and pCdc25C-bound Pin1 with two domains-separated (b), obtained from the modeling using time intervals of 50, 20, and 10 ns (sampling step with 10, 25, and 50, respectively). The total simulation time is 500 ns.

Supplementary Figure 26: Comparison of RMSF values between the simulation and the reconstruction of trajectories for apo Pin1 (a), FFpSPR-bound Pin1 (b), FFpSPR-bound Pin1 (I28A) (c). The trajectories of Pin1 complexes were modeled using time intervals of 20, 8, and 4 ns (sampling step with 10, 25, and 50, respectively). The total simulation time is 200 ns.

Supplementary Figure 27: The distribution of learned edges for the Apo-Pin1 (a), FFpSPR-bound Pin1 (b), and FFpSPR-bound Pin1 (I28A) (c), obtained from the modeling using time intervals of 20, 8, and 4 ns (sampling step with 10, 25, and 50, respectively). The total simulation time is 200 ns.

(iv) I also notice some problems about comparison with other co-variance based approaches that need to be addressed. For example, to compare the energies obtained pairwise interaction of learned edges versus folding stability of alanine mutation of 23 positions, the bound simulation model is used. That does not make sense, instead Apo (unbound model) should be used if they want to compare the folding stability. As another note, the computed pairwise energies are not “free energies” they are completely empirical so they should not call them free energies. They also need to show how their approach give such a strong (but do not make sense as they use an bound state) correlation versus other method did not. They should detail which interactions were missed in the other method.

Response: We double checked our simulations of alanine mutation of 23 positions. We actually used the unbound model to study the folding stability of alanine mutation. We might not have described it clearly. We emphasized it in the revised version.

It is an excellent point that using “free energies” is not rigorous. Based on your suggestion, we changed the “free energies” to “free energy score” in our study to reflect the empirical nature of the function. In addition, we detailed the interactions missed in the other methods for the three case studies. We calculated the suboptimal paths from the WW domain to the catalytic loop based on the covariance matrix obtained from the DCI, Hessian, and CNA methods. We also calculated the network node centralities to describe the distribution of the edges in the protein networks (Supplementary Fig. 33-34). These details can be found in the **Results section (Pages 15-16)** under “Performance comparison between the NRI model and three covariance-based models”, and the **Results section (Pages 16-18)** under “NRI model outperforms three covariance-based tools in capturing long-range interaction”, as well as in the following text:

Computed free energy scores from the NRI modeling agree very well with experimental energies

Based on the ensemble-based perturbation approach, constraint network analysis^{35,36} (CNA) is applied to the ensembles of network topologies generated from MD trajectories for calculating the neighbor stability maps. The stability maps reflect the local stabilities of the residue-residue contacts. Hence, the contribution on free energy due to noncovalent bonding can be estimated by accumulating over the contacts in the stability map (see Supplementary Method 3 for details). This estimation cannot provide absolute free energy values, but it can show its statistical trend. Hence, we call it “free energy score”. ...

To understand the differences of predicted interactions between methods, we identified suboptimal pathways and compared the corresponding node centralities in the pathways calculated based on the covariance matrices obtained from CNA method and the NRI model (Supplementary Figs. 33-34). The node centrality heatmap in the folding stability study shows that the interactions obtained by the CNA method miss the interactions between the WW domain and the α 1-3 helices, and almost all interactions concentrate between the WW domain and the

PPIase core. The NRI model learning yields significantly different results. In particular, the edges between the WW domain and the $\alpha 1$ helix are the main interactions in the alanine mutation of positions 7, 14, 23, 25, and 29. Besides, for the structures whose alanine mutations do not affect the structural stability, the interactions are distributed from the WW domain to the $\alpha 1$ -3 helices or the PPIase core. Because the NRI model can capture the interaction pattern changes corresponding to different structural stabilities resulting from mutations, the accuracy of free energy score estimated by the NRI model dramatically improved over the CNA approach and MD-based method.

NRI model outperforms three covariance-based tools in capturing long-range interaction

Finally, we compared the allosteric pathways learned from the NRI model and the other methods. For the FFpSPR-Pin1 complex in the Pin1 case study, unlike the result of the NRI model, the distributions of edges learned from DCI, Hessian, and CNA methods do not contain the expected interactions between the WW domain and the $\alpha 1$ -3 helices (Supplementary Figs. 40 and 41). Interestingly, D112A/N and C113S caused a considerable reduction in the catalytic activity. E100D, E104K, S105F, S106*, and D112N have been reported as somatic mutations in cancer^{23, 38}. Thus, mutating these residues in the $\alpha 1$ -3 helices may cause chemical shift perturbations in the interaction between the catalytic loop and the PPIase domain. The NRI model successfully learns the interaction patterns in the $\alpha 2$ -3 helices that are ignored by the other three methods. Similarly, upon the pCdc25C binding to the WW domain of the closed conformation, the edges learned from the NRI model are distributed between the WW domain and the helices in the PPIase domain; other methods tend to capture the interaction from the WW domain to the PPIase core. For the extended conformation of FFpSPR-Pin1, the NRI model, Hessian, and CNA methods all capture the high node centralities in the PPIase core. In contrast, the DCI method does not capture the topology from the WW domain to the PPIase core. This suggests that the NRI model is more consistent with the positively and negatively controlled allosteric regulation than other methods.

For the SOD1 WT case study, in contrast to the NRI model, the Hessian and CNA methods ignored the pathways from the DL, DiL, and ZL loops to the EL loop; DCI method missed the pathway from the DL and DiL to the EL loop (Supplementary Figs. 42 and 43). This contradicts the observation that the long active loop (contained by a dimerization loop, disulfide loop, and a zinc-binding loop) is crucial in stabilizing Zn^{2+} binding to the active sites²⁶. For the MEK1 case study, the distribution of edges learned from the NRI model and the other three methods are similar (Supplementary Figs. 44 and 45). Except the pathways directly starting from the activation segment to the αC helix in the S218Sp/S222Sp MEK1, the NRI model determines that the proline-rich loop also plays a bridging role in this message passing. Notably, the proline-rich loop activates the downstream extracellular signal-regulated kinases (ERKs) in cells^{32, 33}.

Supplementary Figure 33: The node centralities in the allosteric pathways between the WW domain and the catalytic loop for the WT and 23 Ala-mutants of unbound Pin1, which are calculated based on the covariance matrices obtained from constraint network analysis (CNA) (**a**) and the NRI model (**b**). Mutations that destabilize more than 3 kcal/mol are shown as red numbers, more than 1 kcal/mol and less than 3 kcal/mol are shown as blue numbers, and less than 1 kcal/mol are shown as black numbers. The color scale bar (from 1.0->0.0) represents the decrease in the importance of a residue measured by normalized node centrality (i.e., the fraction of suboptimal paths going through the node or residue).

Supplementary Figure 40: The node centralities in the allosteric pathways between the WW domain and the catalytic loop for FFpSPR-bound Pin1 (closed conformation), pCdc25C-bound Pin1 (closed conformation), and FFpSPR-bound Pin1 (open conformation). The node centrality is calculated based on the covariance matrices obtained from DCI method, Hessian matrix, CNA method, and the NRI model, respectively. The color scale bar (from 1.0->0.0) represents the decrease in the importance of a residue measured by normalized node centrality (i.e., the fraction of suboptimal paths going through the node or residue).

Supplementary Figure 41: The allosteric pathways mapped on the Pin1 structures for FFpSPR-bound Pin1 (closed conformation), pCdc25C-bound Pin1 (closed conformation), and FFpSPR-bound Pin1 (open conformation) obtained by DCI method (a), Hessian matrix (b), CNA method (c), and the NRI model (d). The covariance matrices obtained from DCI method (e), Hessian matrix (f), CNA method (g), and the NRI model (h).

Supplementary Figure 42: The node centralities in the allosteric pathways between residue G93A and the electrostatic loop for WT- and G93A-SOD1. The node centrality is calculated based on the covariance matrices obtained from DCI method, Hessian matrix, CNA method, and the NRI model.

Supplementary Figure 43: The allosteric pathways mapped on the SOD1 structures for WT- and G93A-SOD1 obtained by DCI method (a), Hessian matrix (b), CNA method (c), and the NRI model (d). The covariance matrices obtained from DCI method (e), Hessian matrix (f), CNA method (g), and the NRI model (h).

Supplementary Figure 44: The node centralities in the allosteric pathways between the activation segment and the α C-helix/proline-rich loop for S218Sp/S222Sp- and E203K-MEK1 complexes. The node centrality is calculated based on the covariance matrices obtained from DCI method, Hessian matrix, CNA method, and the NRI model.

Supplementary Figure 45: The allosteric pathways mapped on the MEK1 structures for S218Sp/S222Sp- and E203K-MEK1 complexes, obtained by DCI method (**a**), Hessian matrix (**b**), CNA method (**c**), and the NRI model (**d**). The covariance matrices obtained from DCI method (**e**), Hessian matrix (**f**), CNA method (**g**), and the NRI model (**h**).

(v) As a minor point, claiming determining the long-range communication with their approach in 50 ns versus some other approach 100 ns does not show an improvement, in MD simulations scale, particularly with GPU processes the time to simulate 50 ns vs 100 ns differ only couple hours and then I think NRI models takes more time to compute than the other methods compared.

Response: It is true that the current NRI models take more time to compute than the other methods compared. The strength of the NRI model is not the computing time but rather its potential to identify some long-range interactions that other methods may miss. This does not mean a replacement of other methods, but our method is complementary to them. We added some discussion in the revised version (**Page 21**).

Reviewer #2 (Remarks to the Author):

I would like to thank the authors for providing a thorough response and careful revise of the draft. The responses from the authors resolve most of my concerns, especially adding a new application for the generator, i.e., evaluation of the difference of relative free energy. Newly added experiments show that the proposed NRI model outperforms the traditional positional covariance-based approaches in terms of the proposed metrics. The clarification on the model part also helps improve the readability of the paper significantly.

From the machine learning perspective, the only concern I have now is about the importance of learning the interaction versus not for NRI in molecular dynamics (MD) simulations. In particular, it would be great to add one VAE baseline which does not have latent variables over edges. Instead, its latent variable per time step would be a Gaussian vector (the latent dimension is a hyperparameter) per node. This can be easily achieved by replacing the softmax readout (over the edge) of the current encoder as Gaussian readout (over the node) like what the original VAE does. The prior would accordingly become the independent standard Gaussian per node. The GNN based encoder and decoder will be both operated on fully connected graphs. The decoder readout layer would still be Gaussian per node. By doing so, we remove the latent interaction graph and instead use fully connected one. Therefore, it could help understand how important it is to learn the interaction for the purpose of generating faithful MD trajectories.

I agree that having a NRI to learn the interactions would greatly improve the interpretability. But it is unclear whether it is necessary for the MD simulation task. It could be the case that without explicit modeling of interactions, VAEs could get better MD simulations. Therefore, comparing with such a baseline would be more convincing.

Response: Thank you for the constructive suggestion, which indeed is an excellent way to illustrate the role of the NRI model. Per your suggestion, we compared the proposed NRI model and the VAE baseline without latent variables over edges. We added some text in the **Result section on Page 15** for details, as follows:

From the methods perspective, modeling the edges explicitly is vital in the NRI architecture. To test the role of graph neural network in NRI, we performed an ablation test. We compared the proposed model and a variational autoencoder (VAE) baseline without latent variables over edges. After splitting the trajectories into training/validating/testing, the MSE results of both models on Pin1, MEK1, and SOD1 are shown in Supplementary Fig. 32 and Supplementary Table 5. We can see the latent variables over the edges can improve the model's performance, and the proposed architecture provides a better framework for modeling edges (residue interactions) of the MD trajectories than other methods.

Supplementary Figure 32: Comparison of mean squared error (MSE) values between the proposed model and the model without latent variables on edges.

Supplementary Table 5: Mean squared error (MSE) values for the proposed model and the model without latent variables on edges.

	Apo-Pin1-3TDB	WT-MEK1	WT-SOD1
Proposed	0.00478	0.00176	0.00430
No latent variables on edges	0.00498	0.00189	0.01111

Reviewers' Comments:

Reviewer #1:

Remarks to the Author:

In this new version, the authors addressed the issues

Reviewer #2:

Remarks to the Author:

Thanks for the response. My concerns have been mostly addressed. Especially, the additional results on the suggested experiments help verify the usefulness of NRI. But it would be great to at least comment on the results. For example, could you comment on why the gap between your method the baseline is much larger on WT-SOD1 than the gaps on the two datasets in Fig. 32?

REVIEWER COMMENTS

Reviewer #1 (Remarks to the Author):

In this new version, the authors addressed the issues

Response: We thank the reviewer for the positive comment on our manuscript.

Reviewer #2 (Remarks to the Author):

Thanks for the response. My concerns have been mostly addressed. Especially, the additional results on the suggested experiments help verify the usefulness of NRI. But it would be great to at least comment on the results. For example, could you comment on why the gap between your method the baseline is much larger on WT-SOD1 than the gaps on the two datasets in Fig. 32?

Response: Thanks for your insightful comment. We added some discussion on page 9 in the paper regarding why the gap between our method and the baseline is much more prominent on WT-SOD1 than on the Apo-Pin1 and WT-MEK1 systems. Supplementary Fig. 28 shows more intensive node-node interactions in the WT-SOD1 than the other two systems. Hence, the effects of graph neural network in NRI using node-node interactions over the VAE baseline (which does not consider node-node interactions) are stronger in WT-SOD1 than in the Apo-Pin1 and WT-MEK1 systems. As a result, compared to the Pin1 and MEK1 systems, the edges learned from three repeated trajectories of the SOD1 system exhibit higher consistency, indicating that the NRI model is more accurate for capturing edges in the WT-SOD1 case (Supplementary Fig. 28). This may cause a more remarkable improvement over the baseline in terms of mean squared error for WT-SOD1 than for the other two systems (Supplementary Fig. 32). It also suggests that the NRI model with latent variables over edges exhibits more significant advantages in more densely interacting systems.